# Discretely Beyond $1/e$: Guided Combinatorial Algorithms for Submodular Maximization

**Yixin Chen, Ankur Nath, Chunli Peng, Alan Kuhnle**
Department of Computer Science & Engineering
Texas A&M University
CollITD Station, TX
{chen777, anath, chunli.peng, kuhnle}@tamu.edu

## Abstract

For constrained, not necessarily monotone submodular maximization, all known approximation algorithms with ratio greater than $1/e$ require continuous ideas, such as queries to the multilinear extension of a submodular function and its gradient, which are typically expensive to simulate with the original set function. For combinatorial algorithms, the best known approximation ratios for both size and matroid constraint are obtained by a simple randomized greedy algorithm of Buchbinder et al. [9]: $1/e \approx 0.367$ for size constraint and $0.281$ for the matroid constraint in $\mathcal{O}(kn)$ queries, where $k$ is the rank of the matroid. In this work, we develop the first combinatorial algorithms to break the $1/e$ barrier: we obtain approximation ratio of $0.385$ in $\mathcal{O}(kn)$ queries to the submodular set function for size constraint, and $0.305$ for a general matroid constraint. These are achieved by guiding the randomized greedy algorithm with a fast local search algorithm. Further, we develop deterministic versions of these algorithms, maintaining the same ratio and asymptotic time complexity. Finally, we develop a deterministic, nearly linear time algorithm with ratio $0.377$.

## 1 Introduction

A nonnegative set function $f : 2^{\mathcal{U}} \to \mathbb{R}^{+}$ is *submodular* iff for all $S \subseteq T \subseteq \mathcal{U}$, $x \in \mathcal{U} \setminus T$, $f(S \cup \{x\}) - f(S) \geq f(T \cup \{x\}) - f(T)$; and $f$ is monotone iff $f(S) \leq f(T)$ for all $S \subseteq T \subseteq \mathcal{U}$. Submodular optimization plays an important role in data science and machine learning [3], particularly in tasks that involve selecting a representative subset of data or features. Its diminishing returns property makes it ideal for scenarios where the incremental benefit of adding an element to a set decreases as the set grows. Applications include sensor placement for environmental monitoring [19, 27], where the goal is to maximize coverage with limited sensors, feature selection [22, 18, 2] in machine learning to improve model performance and reduce overfitting, and data summarization [23, 31] for creating concise and informative summaries of large datasets. Further, many of these applications employ submodular objective functions that are non-monotone, *e.g.* Mirzasoleiman et al. [23], Tschiatschek et al. [31]. Formally, we study the optimization problem (SM): $\max f(S)$, *s.t.* $S \in \mathcal{I}$, where $f$ is nonnegative, submodular and not necessarily monotone; and $\mathcal{I} \subseteq 2^{\mathcal{U}}$ is a family of feasible subsets. Specifically, we consider two cases: when $\mathcal{I}$ is a size constraint (all sets of size at most $k$); and more generally, when $\mathcal{I}$ is an arbitrary matroid of rank $k$.

In this field, algorithms typically assume access to a *value oracle* for the submodular function $f$, and the efficiency of an algorithm is measured by the number of queries to the oracle, because evaluation of the submodular function is typically expensive and dominates other parts of the computation. In the general, not necessarily monotone case, the approximability of constrained submodular optimization in the value oracle model is not well understood. For several years, $1/e \approx 0.367$ was conjectured

Table 1: The prior state-of-the-art and the ratios achieved in this paper, in each category: deterministic (det), randomized combinatorial (cmb), and continuous (cts).

| Constraint | Reference | Query | Ratio | Type |
|---|---|---|---|---|
| Size | Buchbinder and Feldman [5] | $\mathcal{O}\left(k^3 n\right)$ | $1/e \approx 0.367$ | Det |
| | Buchbinder et al. [9] | $\mathcal{O}\left(kn\right)$ | $1/e$ | Cmb |
| | Buchbinder and Feldman [7] | $\mathrm{poly}(n)$ | $0.401$ | Cts |
| | Algorithm 2 | $\mathcal{O}\left(kn/\varepsilon\right)$ | $0.385 - \varepsilon$ | Cmb |
| | Algorithm 11 | $\mathcal{O}\left(kn\left(\frac{10}{9\varepsilon}\right)^{\frac{20}{9\varepsilon}-1}\right)$ | $0.385 - \varepsilon$ | Det |
| | Algorithm 14 | $\mathcal{O}\left(\log(k)n\left(\frac{10}{3\varepsilon}\right)^{\frac{20}{3\varepsilon}}\left(\frac{5}{\varepsilon}\right)^{\frac{10}{\varepsilon}-1}\right)$ | $0.377 - \varepsilon$ | Det |
| Matroid | Sun et al. [30] | $\mathcal{O}\left(k^2 n^2\right)$ | $0.283 - \mathcal{O}\left(\frac{1}{k^2}\right)$ | Det |
| | Buchbinder et al. [9] | $\mathcal{O}\left(kn\right)$ | $0.283 - \varepsilon$ | Cmb |
| | Buchbinder and Feldman [7] | $\mathrm{poly}(n)$ | $0.401$ | Cts |
| | Algorithm 2 | $\mathcal{O}\left(kn/\varepsilon\right)$ | $0.305 - \varepsilon$ | Cmb |
| | Algorithm 11 | $\mathcal{O}\left(kn\left(\frac{10}{9\varepsilon}\right)^{\frac{20}{9\varepsilon}-1}\right)$ | $0.305 - \varepsilon$ | Det |

to be the best ratio, as this ratio is obtained by the measured continuous greedy [15] algorithm that also gets the $1 - 1/e$ ratio in the monotone setting, which is known to be optimal [24]. However, in several landmark works, the $1/e$ barrier was broken: first to $0.371$ by Buchbinder et al. [9] (for size constraint only) and subsequently to $0.372$ by Ene and Nguyen [13], then $0.385$ by Buchbinder and Feldman [6]. Very recently, the best known approximation factor has been improved to $0.401$ [7]. On the other hand, the best hardness result is $0.478$ [16, 28].

All of the algorithms improving on the $1/e$ ratio use oracle queries to the *multilinear extension* of a submodular function and its gradient. The multilinear extension relaxes the submodular set function to allow choosing an element with probability in $[0, 1]$. Although this is a powerful technique, the multilinear extension must be approximated by polynomially many random samples of the original set function oracle. Unfortunately, this leads to a high query complexity for these algorithms, which we term *continuous algorithms*; typically, the query complexity to the original submodular function is left uncomputed. As an illustration, we compute in Appendix B that the continuous algorithm of Buchbinder and Feldman [6] achieves ratio of $0.385$ with query complexity of $\mathcal{O}\left(n^{11} \log(n)\right)$ to the set function oracle. Consequently, these algorithms are of mostly theoretical interest – the time cost of running on tiny instances (say, $n < 100$) is already prohibitive, as demonstrated by Chen and Kuhnle [12] where a continuous algorithm required more than $10^9$ queries to the set function on an instance with $n = 87, k = 10$.

For size and matroid constraints, the current state-of-the-art approximation ratio for a combinatorial algorithm (*i.e.* not continuous) is obtained by the RANDOMGREEDY algorithm (Algorithm 1) of Buchbinder et al. [9]. RANDOMGREEDY achieves ratio $1/e \approx 0.367$ for size constraint, and $0.283 - \varepsilon$ ratio for matroid constraint; its query complexity is $\mathcal{O}\left(kn\right)$. Thus, there is no known combinatorial algorithm that breaks the $1/e$ barrier; and therefore, no such algorithm is available to be used in practice on any of the applications of SM described above.

Moreover, closing the gap between ratios achieved by deterministic and randomized algorithms for SM has been the focus of a number of recent works [5, 17, 11, 8]. In addition to theoretical interest, deterministic algorithms are desirable in practice, as a ratio that holds in expectation may fail on any given run with constant probability. Buchbinder and Feldman [5] introduced a linear programming method to derandomize the RANDOMGREEDY algorithm (at the

---

**Algorithm 1:** Buchbinder et al. [9]

1 **Procedure** RANDOMGREEDY $(f, k)$:
2      **Input:** oracle $f$, size constraint $k$
3      **Initialize:** $A_0 \leftarrow \emptyset$
4      **for** $i \leftarrow 1$ *to* $k$ **do**
5          $M_i \leftarrow$ $\arg\max_{S \subseteq \mathcal{U}, |S|=k} \sum_{x \in S} \Delta(x|A_{i-1})$
6          $x_i \leftarrow$ a uniformly random element from $M_i$
7          $A_i \leftarrow A_{i-1} + x_i$
8      **end**
9      **return** $A_k$

---

expense of additional time complexity), meaning that the best known ratios for deterministic algorithms are again given by RANDOMGREEDY. There is no known method to derandomize continuous

algorithms, as the only known way to approximate the multilinear extension of a general submodular set function relies on random sampling methods. Özcan et al. [26], however, introduced a deterministic estimation via Taylor series approximation, but this approach is limited to a specific class of submodular functions that can be expressed as weighted compositions of analytic and multilinear functions. Therefore, there is no known deterministic algorithm that breaks the $1/e$ barrier. The best known ratio in each category of continuous, combinatorial, and deterministic algorithms is summarized in Table 1. In this work, we consider the following questions:

*Can combinatorial algorithms, and separately, deterministic algorithms, obtain approximation ratios for* SM *beyond* $1/e$? *If so, are the resulting algorithms practical and do they yield empirical improvements in objective value over existing algorithms?*

## 1.1 Contributions

In this work, we improve the best known ratio for a combinatorial algorithm for size-constrained SM to $0.385 - \varepsilon \approx 1/e + 0.018$. This is achieved by using the result of a novel local search algorithm to guide the RANDOMGREEDY algorithm. Overall, we obtain query complexity of $\mathcal{O}\left(kn/\varepsilon\right)$, which is at worst quadratic in the size of the ground set, since $k \leq n$. Thus, this algorithm is practical and can run on moderate instance sizes; the first algorithm with ratio beyond $1/e$ for which this is possible. Further, we extend this algorithm to the matroid constraint, where it improves the best known ratio of a combinatorial algorithm for a general matroid constraint from $0.283$ of RANDOMGREEDY to $0.305 - \varepsilon$.

Secondly, we obtain these same approximation ratios with deterministic algorithms. The ideas are similar to the randomized case, except we leverage a recently formulated algorithm INTERPOLATEDGREEDY [11] as a replacement for guided RANDOMGREEDY. The analysis of INTERPOLATEDGREEDY has similar recurrences (up to low order terms) and the algorithm can be guided in a similar fashion to RANDOMGREEDY, but is amenable to derandomization. The derandomization only adds a constant factor, albeit one that is exponential in $(1/\varepsilon)$.

Next, we seek to lower the query complexity further, while still improving the $1/e$ ratio. As INTERPOLATEDGREEDY can be sped up to $\mathcal{O}_\varepsilon(n \log k)$, the bottleneck becomes the local search procedure. Thus, we develop a faster way to produce the guiding set $Z$ by exploiting a run of (unguided) INTERPOLATEDGREEDY and demonstrating that a decent guiding set is produced if the algorithm exhibits nearly worst-case behavior. With this method, we achieve a deterministic algorithm with ratio $0.377 \approx 1/e + 0.01$ in $\mathcal{O}_\varepsilon(n \log k)$ query complexity, which is nearly linear in the size of the ground set (since $k = O(n)$).

Finally, we demonstrate the practical utility of our combinatorial $0.385$-approximation algorithm by implementing it and evaluating in the context of two applications of size-constrained SM on moderate instance sizes (up to $n = 10^4$). We evaluate it with parameters set to enforce a ratio $> 1/e$. It outperforms both the standard greedy algorithm and RANDOMGREEDY by a significant margin in terms of objective value; moreover, it uses about twice the queries of RANDOMGREEDY and is orders of magnitude faster than existing local search algorithms.

## 1.2 Additional Related Work

**Derandomization.** Buchbinder and Feldman [5] introduced a linear programming (LP) method to derandomize the RANDOMGREEDY algorithm, thereby obtaining ratio $1/e$ with a deterministic algorithm. Further, Sun et al. [30] were able to apply this technique to RANDOMGREEDY for matroids. A disadvantage of this approach is an increase in the query complexity over the original randomized algorithm. Moreover, we attempted to use this method to derandomize our guided RANDOMGREEDY algorithm, but were unsuccessful. Instead, we obtained our deterministic algorithms by guiding the INTERPOLATEDGREEDY algorithm instead of RANDOMGREEDY; this algorithm is easier to derandomize, notably without increasing the asymptotic query complexity.

**Relationship to Buchbinder and Feldman [6].** The continuous, $0.385$-approximation algorithm of Buchbinder and Feldman [6] guides the measured continuous greedy algorithm using the output of a continuous local search algorithm, in analogous fashion to how we guide RANDOMGREEDY with the output of a combinatorial local search. However, the analysis of RANDOMGREEDY is much different from that of measured continuous greedy, although the resulting approximation factor is the same.

Specifically, Buchbinder and Feldman [6] obtain their ratio by optimizing a linear program mixing the continous local search and guided measured continous greedy; in contrast, we use submodularity and the output of our fast local search to formulate new recurrences for guided RANDOMGREEDY, which we then solve.

**Local search algorithms.** Local search is a technique widely used in combinatorial optimization. Nemhauser et al. [25] introduced a local search algorithm for monotone functions under size constraint; they showed a ratio of $1/2$, but noted that their algorithm may run in exponential time. Subsequently, local search has been found to be useful, especially for non-monotone functions. Feige et al. [14] proposed a $1/3$ approximation algorithm with $\mathcal{O}\left(n^4/\varepsilon\right)$ queries for the unconstrained submodular maximization problem utilizing local search. Meanwhile, Lee et al. [21] proposed a local search algorithm for general SM with matroid constraint, attaining $1/4 - \varepsilon$ approximation ratio with a query complexity of $\mathcal{O}\left(k^5 \log(k)n/\varepsilon\right)$. We propose our own FASTLS in Section 2.1, yielding a ratio of $1/2$ for monotone cases and $1/4$ for non-monotone cases through repeated applications of FASTLS, while running in $\mathcal{O}\left(kn/\varepsilon\right)$ queries.

**Fast approximation algorithms.** Buchbinder et al. [10] developed a faster version of RANDOM-GREEDY for size constraint that reduces the query complexity to $\mathcal{O}_\varepsilon\left(n\right)$ with ratio of $1/e - \varepsilon$. Chen and Kuhnle [11] proposed LINEARCARD, the first deterministic, linear-time algorithm with an $1/11.657$-approximation ratio for size constraints. Also, Han et al. [17] introduced TWINGREEDY, a $0.25$-approximation algorithm with a query complexity of $\mathcal{O}\left(kn\right)$ for matroid constraints. These algorithms are fast enough to be used as building blocks for our FASTLS, which requires as an input a constant-factor approximation in $\mathcal{O}\left(kn\right)$ queries.

**Relationship to Tukan et al. [32].** During the submission of this paper, we noticed an independent and parallel work by Tukan et al. [32], which proposed a different $0.385$-approximation algorithm. Both papers start from a similar idea-guiding the random greedy algorithm with a fast algorithm to find a local optimum. However, Tukan et al. [32] only considered size constraint and focused on algorithm speedup. They introduced a randomized local search algorithm and used its output to guide the stochastic greedy of Buchbinder et al. [9], achieving a query complexity of $\mathcal{O}_\varepsilon\left(n + k^2\right)$. On the other hand, we 1) address a more general constraint-matroid constraint; 2) for size constraint, present an asymptotically faster algorithm that uses a novel way of guiding with partial solutions from random greedy itself, which are not local optima, thereby achieving ratio $0.377 - \varepsilon$ with $\mathcal{O}_\varepsilon\left(n \log(k)\right)$ queries; and 3) derandomize these algorithms.

## 1.3 Preliminaries

**Notation.** In this section, we establish the notations employed throughout the paper. We denote the marginal gain of adding $A$ to $B$ by $\Delta(A|B) = f(A \cup B) - f(B)$. For every set $S \subseteq \mathcal{U}$ and an element $x \in \mathcal{U}$, we denote $S \cup \{x\}$ by $S + x$, and $S \setminus \{x\}$ by $S - x$. Given a constraint and its related feasible sets $\mathcal{I}$, let $O \in \arg\max_{S \in \mathcal{I}} f(S)$; that is, $O$ is an optimal solution. To simplify the pseudocode and the analysis, we add $k$ *dummy elements* into the ground set, where the dummy element serves as a null element with zero marginal gain when added to any set. The symbol $e_0$ is utilized to represent a dummy element.

**Submodularity.** A set function $f : 2^{\mathcal{U}} \to \mathbb{R}^+$ is submodular, if $\Delta(x|S) \geq \Delta(x|T)$ for all $S \subseteq T \subseteq \mathcal{U}$ and $x \in \mathcal{U} \setminus T$, or equivalently, for all $A, B \subseteq \mathcal{U}$, it holds that $f(A) + f(B) \geq f(A \cup B) + f(A \cap B)$.

**Constraints.** In this paper, our focus lies on two constraints: size constraint and matroid constraint. For size constraint, we define the feasible subsets as $\mathcal{I}(k) = \{S \subseteq \mathcal{U} : |S| \leq k\}$, where $k$ is an input parameter. The matroid constraint is defined in Appendix A.

**Organization.** Our randomized algorithms are described in Section 2, with two subroutines, FASTLS and GUIDEDRG, in Section 2.1 and 2.2, respectively. Due to space constraints, we provide only a sketch of the analysis for size constraint in the main text. The full pseudocodes and formal proofs for both size and matroid constraint are provided in Appendix C. Then, we briefly sketch the deterministic approximation algorithms in Section 2.3, with full details provided in Appendix D. Next, we introduce the nearly linear-time deterministic algorithm in Section 3, with omitted analysis provided in Appendix E. Our empirical evaluation is summarized in Section 4. In Section 5, we discuss limitations and future directions.

| **Algorithm 2:** Randomized combinatorial approximation algorithm. |
|---|

1 **Input:** Instance $(f, \mathcal{I})$, a constant-factor approximation $Z_0$, switch time $t \in [0, 1]$, accuracy
   $\varepsilon > 0$
2 $Z \leftarrow \text{FastLS}(f, \mathcal{I}, Z_0, \varepsilon)$                     `/* find local optimum Z */`
3 $A \leftarrow \text{GuidedRG}(f, \mathcal{I}, Z, t)$            `/* guided by local optimum Z */`
4 **return** $\arg\max\{f(Z), f(A)\}$

## 2    A Randomized $(0.385 - \varepsilon)$-approximation in $\mathcal{O}(kn/\varepsilon)$ Queries

In this section, we present our randomized approximation algorithm (Alg. 2) for both size and matroid constraints. This algorithm improves the state-of-the-art, combinatorial approximation ratio to $0.385 - \varepsilon \approx 1/e + 0.018$ for size constraint, and to $0.305 - \varepsilon \approx 0.283 + 0.022$ for matroid constraint.

**Algorithm Overview.** In overview, Alg. 2 consists of two components, which are detailed below. The first component is a local search algorithm, FastLS (Alg. 4 in Appendix C.1), described in detail in Section 2.1. In brief, the local search algorithm takes an accuracy parameter $\varepsilon > 0$ and a constant-factor, approximate solution $Z_0$ as input, which may be produced by any approximation algorithm with better than $\mathcal{O}(kn)$ query complexity. The second component is a random greedy algorithm, GuidedRG (Alg. 6 in Appendix C.2), that is guided by the output $Z$ of the local search, described in detail in Section 2.2. Also, GuidedRG takes a parameter $t \in [0, 1]$, which is the switching time (as fraction of the budget or rank $k$) from guided to unguided behavior. The candidate with best $f$ value from the two subroutines is returned.

If $f(Z) < \alpha\text{OPT}$ (otherwise, there is nothing to show), then the local search set satisfies our definition of $((1 + \varepsilon)\alpha, \alpha)$-guidance set (Def. 2.1 below). Under this guidance, we show that GuidedRG produces a superior solution compared to its unguided counterpart. The two components, FastLS and GuidedRG are described in Sections 2.1 and 2.2, respectively. The following theorem is proven in Section 2.2 (size constraint) and Appendix C.2.2 (matroid constraint).

**Theorem 2.1.** Let $(f, \mathcal{I})$ be an instance of SM. Let $\varepsilon > 0$, and $k \geq 1/\varepsilon$. Algorithm 2 achieves an expected $(0.385 - \varepsilon)$-approximation ratio for size constraint with $t = 0.372$, and an expected $(0.305 - \varepsilon)$-approximation ratio for matroid constraint with $t = 0.559$. The query complexity of the algorithm is $\mathcal{O}(kn/\varepsilon)$.

### 2.1   The Fast Local Search Algorithm

In this section, we introduce FastLS (Alg. 4), which is the same for size or matroid constraints. There are several innovations in FastLS that result in $\mathcal{O}(kn/\varepsilon)$ time complexity, where $k$ is the maximum size of a feasible set, and $\varepsilon > 0$ is an input accuracy parameter.

In overview, the algorithm maintains a feasible set $Z$; initially, $Z = Z_0$, where $Z_0$ is an input set which is a constant approximation to OPT. The value of $Z$ is iteratively improved via swapping, which is done in the following way. For each element $a \in \mathcal{U}$, we compute $\Delta(a|Z \setminus a)$; if $a \notin Z$, this is just the gain of $a$ to $Z$; this requires $\mathcal{O}(n)$ queries. Then, if $a \in Z$ and $e \notin Z$ such that $Z \setminus a + e$ is feasible, and $\Delta(e|Z) - \Delta(a|Z \setminus a) \geq \frac{\varepsilon}{k} f(Z)$, then $a$ is swapped in favor of $e$. If no such swap exists, the algorithm terminates.

One can show that, for each swap, the value of $Z$ increases by at least a multiplicative $(1 + \varepsilon/k)$ factor. Since $f(Z)$ is initialized to a constant fraction of OPT, it follows that we make at most $\mathcal{O}(k/\varepsilon)$ swaps. Since each swap requires $\mathcal{O}(n)$ queries, this yields the query complexity of the algorithm: $\mathcal{O}(kn/\varepsilon)$. In addition, if $f$ is monotone, FastLS gets ratio of nearly $1/2$ for FastLS. A second repetition of FastLS yields a ratio of $1/4$ in the case of general (non-monotone) $f$, as shown in Appendix C.1.2. Thus, FastLS may be of independent interest, as local search obtains good objective values empirically and is commonly used in applications.

For our purposes, we want to use the output of FastLS to guide RandomGreedy. Since we will also use another algorithm for a similar purpose in Section 3, we abstract the properties needed for such a guidance set. Intuitively, a set $Z$ is a good guidance set if it has a low $f$-value and also ensures that the value of its intersection and union with an optimal solution are poor.

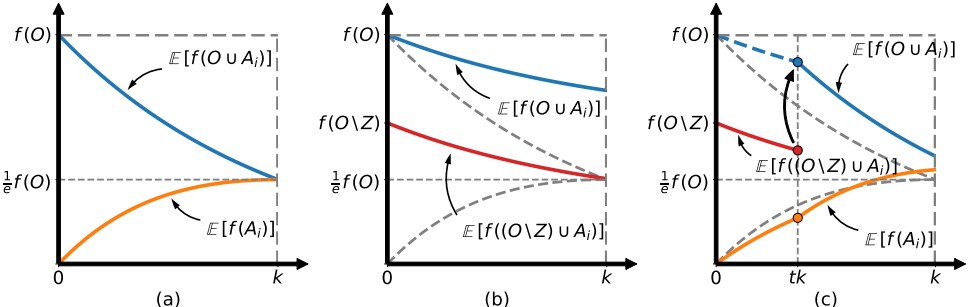

Figure 1: **(a)**: The evolution of $\mathbb{E}\left[f\left(O \cup A_i\right)\right]$ and $\mathbb{E}\left[f\left(A_i\right)\right]$ in the worst case of the analysis of RANDOMGREEDY, as the partial solution size increases to $k$. **(b)**: Illustration of how the degradation of $\mathbb{E}\left[f\left(O \cup A_i\right)\right]$ changes as we introduce an $(0.385 + \varepsilon, 0.385)$-guidance set. **(c)**: The updated degradation with a switch point $tk$, where the algorithm starts with guidance and then switches to running without guidance. The dashed curved lines depict the unguided values from **(a)**.

**Definition 2.1.** A set $Z$ is a $(\alpha, \beta)$-guidance set, if given constants $\alpha, \beta \in (0, 0.5)$ and optimum solution $O$, it holds that: 1) $f(Z) < \alpha f(O)$; 2) $f(O \cap Z) \le \alpha f(O)$; 3) $f(O \cup Z) \le \beta f(O)$, or alternatively, 3') $f(O \cap Z) + f(O \cup Z) \le (\alpha + \beta) f(O)$.

Lemma 2.2 (proved in Appendix C.1.1) implies that for the FASTLS output $Z$, if $f(Z) < \alpha$OPT, then $Z$ is $((1 + \varepsilon)\alpha, \alpha)$-guidance set.

**Lemma 2.2.** Let $\varepsilon > 0$, and let $(f, \mathcal{I}(\mathcal{M}))$ be an instance of SM. The input set $Z_0$ is an $\alpha_0$-approximate solution to $(f, \mathcal{I}(\mathcal{M}))$. FASTLS (Alg. 4) returns a solution $Z$ with $\mathcal{O}\left(kn \log(1/\alpha_0)/\varepsilon\right)$ queries such that $f(S \cup Z) + f(S \cap Z) < (2 + \varepsilon) f(Z)$, where $S \in \mathcal{I}(\mathcal{M})$.

## 2.2 Guiding the RANDOMGREEDY Algorithm

In this section, we discuss the guided RANDOMGREEDY algorithm (Alg. 6) using an $((1 + \varepsilon)\alpha, \alpha)$-guidance set $Z$ returned by FASTLS. Due to space constraints, we only consider the size constraint in the main text. The ideas for the matroid constraint are similar, although the final recurrences obtained differ. The version for matroid constraints is given in Appendix C.2.2.

The algorithm GUIDEDRG is simple to describe: it maintains a partial solution $A$, initially empty. It takes as parameters the switching time $t$ and guidance set $Z$. While the partial solution satisfies $|A| < tk$, the algorithm operates as RANDOMGREEDY with ground set $\mathcal{U} \setminus Z$; after $|A| \ge tk$, it operates as RANDOMGREEDY with ground set $\mathcal{U}$. Pseudocode is provided in Appendix C.2.

**Overview of analysis.** For clarity, we first describe the (unguided) RANDOMGREEDY analysis from Buchbinder et al. [9]. There are two recurrences: the first is the greedy gain:

$$\mathbb{E}\left[f\left(A_i\right) - f\left(A_{i-1}\right)\right] \ge \frac{1}{k}\mathbb{E}\left[f\left(O \cup A_{i-1}\right) - f\left(A_{i-1}\right)\right].$$

Intuitively, the gain at iteration $i$ is at least a $1/k$ fraction of the difference between $f(O \cup A)$ and $A$, in expectation, where $A$ is the partial solution. If $f$ were monotone, the right hand side would be at least $(\text{OPT} - f(A))/k$. However, in the case that $f$ is not monotone, the set $O \cup A$ may have value smaller than OPT.

To handle this case, it can be shown that the expected value of $f(O \cup A)$ satisfies a second recurrence:

$$\mathbb{E}\left[f\left(O \cup A_i\right)\right] \overset{(a)}{\ge} \left(1 - \frac{1}{k}\right)\mathbb{E}\left[f\left(O \cup A_{i-1}\right)\right] + \frac{1}{k}\mathbb{E}\left[f\left(O \cup A_{i-1} \cup M_i\right)\right] \overset{(b)}{\ge} \left(1 - \frac{1}{k}\right)\mathbb{E}\left[f\left(O \cup A_{i-1}\right)\right],$$

where $M_i$ is the set of elements with the top $k$ marginal gains at iteration $i$, *(a)* is from submodularity, and *(b)* is from nonnegativity. Thus, this expected value, while initially OPT (since $A_0 = \emptyset$), may degrade but is bounded.

Both of these recurrences are solved together to prove the expected ratio of $1/e$ for RANDOMGREEDY: the worst-case evolution of the expected values of $f(A_i)$, $f(O \cup A_i)$, according to this analysis,

is illustrated in Fig. 1(a). Observe that $f(A_i)$ converges to OPT$/e$ (as required for the ratio), and *observe that $f(O \cup A_i)$ also converges to OPT$/e$*. Thus, very little gain is obtained in the later stages of the algorithm, as illustrated in the plot. The overarching idea of the guided version of the algorithm is to obtain a better degradation of $\mathbb{E}\left[f\left(O \cup A_i\right)\right]$, leading to better gains later in the algorithm that improve the worst-case ratio. In the following, we elaborate on this goal, the achievement of which is illustrated in Fig. 1(c).

**Stage 1: Recurrences when avoiding $Z$.** Suppose $Z$ is an $(\alpha, \beta)$-guidance set, and that RANDOM-GREEDY selects elements as before, but excluding $Z$ from the ground set. Then, the recurrences change as follows. The recurrence for the gain becomes:

$$\mathbb{E}\left[f\left(A_i\right) - f\left(A_{i-1}\right)\right] \geq \frac{1}{k}\mathbb{E}\left[f\left(\left(O \setminus Z\right) \cup A_{i-1}\right) - f\left(A_{i-1}\right)\right], \tag{1}$$

where $O \setminus Z$ replaces $O$ since we select elements outside of the set $Z$. For the second recurrence, we can lower bound the term $\mathbb{E}\left[f\left(O \cup A_{i-1} \cup M_i\right)\right]$ using submodularity and the fact that $Z \cap A_{i-1} = \emptyset$:

$$\mathbb{E}\left[f\left(O \cup A_i\right)\right] \geq \left(1 - \frac{1}{k}\right)\mathbb{E}\left[f\left(O \cup A_{i-1}\right)\right] + \frac{1}{k}\mathbb{E}\left[f\left(O\right) - f\left(O \cup Z\right)\right]. \tag{2}$$

Finally, a similar recurrence to (2) also holds for $f\left(\left(O \setminus Z\right) \cup A_i\right)$; both are needed for the analysis. Since $Z$ is a guidance set, by submodularity, $f(O \setminus Z) \geq f(O) - f(O \cap Z) \geq (1 - \alpha)$OPT, which ensures that some gain is available by selection outside of $Z$. And $f(O) - f(O \cup Z) \geq (1 - \beta)$OPT, which means that the degradation recurrences are improved.

The blue line in Figure 1(b) depicts this improved degradation with the size of the partial solution. However, this improvement comes at a cost: a smaller increase in $\mathbb{E}\left[f\left(A_i\right)\right]$ is obtained over the unguided version. Therefore, to obtain an improved ratio we switch back to the regular behavior of RANDOMGREEDY – intuitively, this shifts the relatively good, earlier behavior of RANDOMGREEDY to later in the algorithm.

**Stage 2: Switching back to selection from whole ground set.** After the switch, the recurrences revert back to the original ones, but with different starting values. Since $\mathbb{E}\left[f\left(O \cup A_i\right)\right]$ was significantly enhanced in the first stage, in the final analysis we get an overall improvement over the unguided version. The blue line in Figure 1(c) demonstrates the degradation of $\mathbb{E}\left[f\left(O \cup A_i\right)\right]$ over two stages, while the orange line depicts how the approximation ratio converges to a value $0.385 > 1/e$.

The above analysis sketch can be formalized and the resulting recurrences solved: the results are stated in the following lemma, which is formally proven in Appendix C.2.1.

**Lemma 2.3.** With an input size constraint $\mathcal{I}$ and a $((1 + \varepsilon)\alpha, \alpha)$-guidance set $Z$, GUIDE-DRG returns set $A_k$ with $\mathcal{O}\left(kn\right)$ queries, *s.t.* $\mathbb{E}\left[f\left(A_k\right)\right] \geq \left[\left(2 - t - \frac{1}{k}\right)\left(1 - \frac{1}{k}\right)e^{t-1} - e^{-1}\right.$ $-(1 + \varepsilon)\alpha\left(\left(1 - \frac{1}{k}\right)^2 e^{t-1} - e^{-1}\right) - \alpha\left(\left(1 + \frac{1-t}{1-\frac{1}{k}}\right)e^{t-1} - \left(2 - \frac{1}{k}\right)e^{-1}\right)\right]f\left(O\right).$

From Lemma 2.3, we can directly prove the main result for size constraint.

*Proof of Theorem 2.1 under size constraint.* Let $(f, \mathcal{I})$ be an instance of SM, with optimal solution set $O$. If $f(Z) \geq (0.385 - \varepsilon)f(O)$ under size constraint, the approximation ratio holds immediately. Otherwise, by Lemma 2.2, FASTLS returns a set $Z$ which is an $((1 + \varepsilon)\alpha, \alpha)$-guidance set, where $\alpha = 0.385 - \varepsilon$. By Lemma 2.3,

$$\mathbb{E}\left[f\left(A_k\right)\right] \geq \left[(2 - t - \varepsilon)(1 - \varepsilon)e^{t-1} - e^{-1} - (0.385 - 0.615\varepsilon)\left((1 - \varepsilon)^2 e^{t-1} - e^{-1}\right)\right.$$

$$\left.-(0.385 - \varepsilon)\left(\left(1 + \frac{1-t}{1-\varepsilon}\right)e^{t-1} - (2 - \varepsilon)e^{-1}\right)\right]f\left(O\right) \qquad\qquad (\forall k \geq \tfrac{1}{\varepsilon})$$

$$\geq (0.385 - \varepsilon)f\left(O\right). \qquad\qquad (t = 0.372)$$

$\square$

## 2.3  Deterministic approximation algorithms

In this section, we outline the deterministic algorithms, for size and matroid constraints. The main idea is similar, but we replace GUIDEDRG with a deterministic subroutine. For simplicity, we

present a randomized version in Appendix D.2 as Alg. 10, which we then derandomize (Alg. 11 in Appendix D.3). Further discussion is provided in Appendix D.

**Algorithm overview.** Chen and Kuhnle [11] proposed a randomized algorithm, INTERPOLATED-GREEDY, which may be thought of as an interpolation between standard greedy [25] and RANDOM-GREEDY [9]. Instead of picking $k$ elements, each randomly chosen from the top $k$ marginal gains, it picks $\ell = \mathcal{O}(1/\varepsilon)$ sets randomly from $\mathcal{O}(\ell)$ candidates. Although it uses only a constant number of rounds, the recurrences for INTERPOLATEDGREEDY are similar to the RANDOMGREEDY ones discussed above, so we can guide it similarly.

To select the candidate sets in each iteration, we replace INTERLACEGREEDY (the subroutine of INTERPOLATEDGREEDY proposed in Chen and Kuhnle [11]) with a guided version: GUIDEDIG-S (Alg. 9 in Appendix D.1.1) for size constraint, and GUIDEDIG-M (Alg. 8 in Appendix D.1) for matroid constraint. Since only $\ell$ random choices are made, each from $\mathcal{O}(\ell)$ sets, there are at most $\mathcal{O}\left(\ell^{\mathcal{O}(\ell)}\right)$ possible solutions, where $\ell$ is a constant number depending on $\varepsilon$. Notably, we are still able to obtain the same approximation factors as in Section 2. The proof of Theorem 2.4 is provided in Appendices D.2 and D.3.

**Theorem 2.4.** Let $(f, k)$ be an instance of SM, with the optimal solution set $O$. Alg. 11 achieves a deterministic $(0.385 - \varepsilon)$ approximation ratio with $t = 0.372$, and a deterministic $(0.305 - \varepsilon)$ approximation ratio with $t = 0.559$. The query complexity of the algorithm is $\mathcal{O}\left(kn\ell^{2\ell-1}\right)$ where $\ell = \frac{10}{9\varepsilon}$.

## 3   Deterministic Algorithm with Nearly Linear Query Complexity

In this section, we sketch a deterministic algorithm with $(0.377 - \varepsilon)$ approximation ratio and $\mathcal{O}_{\varepsilon}(n \log(k))$ query complexity for the size constraint. A full pseudocode (Alg. 14) and analysis is provided in Appendix E.

**Description of algorithm.** Our goal is to improve the asymptotic $\mathcal{O}_{\varepsilon}(kn)$ query complexity. Recall that in Section 2, we described a deterministic algorithm that employed the output of local search to guide the INTERPOLATEDGREEDY algorithm, which obeys similar recurrences to RANDOMGREEDY. To produce the $\ell$ candidate sets for each iteration of INTERPOLATEDGREEDY, a greedy algorithm (guided INTERLACEGREEDY) is used. These algorithms can be sped up using a descending thresholds technique. This results in THRESHGUIDEDIG (Alg. 12 in Appendix E.1), which achieves $\mathcal{O}_{\varepsilon}(n \log k)$ query complexity for the guided part of our algorithm.

However, the local search FASTLS still requires $\mathcal{O}(kn/\varepsilon)$ queries, so we seek to find a guidance set in a faster way. Recall that, in the definition of guidance set $Z$, the value $f(Z)$ needs to dominate both $f(O \cap Z)$ and $f(O \cup Z)$. To achieve this with faster query complexity, we employ a run of unguided INTERPOLATEDGREEDY. Consider the recurrences plotted in Fig. 1(a) – if the worst-case degradation occurs, then at some point the value of $f(A_i)$ becomes close to $f(O \cup A_i)$. On the other hand, if the worst-case degradation does not occur, then the approximation factor of INTERPOLATEDGREEDY is improved (see Fig. 2). Moreover, if we ensure that at every stage, $A_i$ contains no elements that contribute a negative gain, then we will also have $f(A_i) \geq f(O \cap A_i)$.

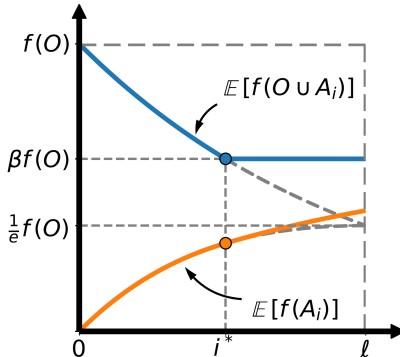

Figure 2: Depiction of how analysis of INTERPOLATEDGREEDY changes if there is no $(0.377, 0.46)$-guidance set.

To execute this idea, we run (derandomized, unguided) INTERPOLATEDGREEDY, and consider all $\mathcal{O}\left(\ell^{\ell}\right)$ intermediate solutions. Each one of these is pruned (by which we mean, any element with negative contribution is discarded until none such remain). Then, the guided part of our algorithm executes with every possible candidate intermediate solution as the guiding set; finally, the feasible set encountered with maximum $f$ value is returned.

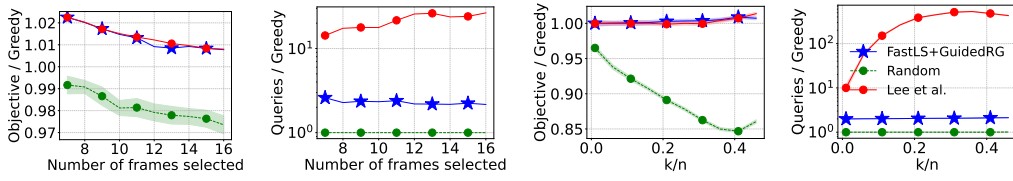

(a) Video Summarization, solution value

(b) Video Summarization, queries

(c) Maximum Cut (ER), solution value

(d) Maximum Cut (ER), queries

Figure 3: The objective value (higher is better) and the number of queries (log scale, lower is better) are normalized by those of STANDARDGREEDY. Our algorithm (blue star) outperforms every baseline on at least one of these two metrics.

The tradeoff between the first and second parts of the algorithm is optimized with $\alpha = 0.377$ and $\beta = 0.46$. That is, if INTERPOLATEDGREEDY produces an $(\alpha, \beta)$-guidance set, the guided part of our algorithm achieves ratio 0.377; otherwise, INTERPOLATEDGREEDY has ratio at least 0.377. We have the following theorem. The algorithms and analysis sketched here are formalized in Appendix E.

**Theorem 3.1.** Let $(f, k)$ be an instance of SM, with the optimal solution set $O$. Algorithm 14 achieves a deterministic $(0.377 - \varepsilon)$ approximation ratio with $\mathcal{O}(n \log(k){\ell_1}^{2\ell_1}{\ell_2}^{2\ell_2 - 1})$ queries, where $\ell_1 = \frac{10}{3\varepsilon}$ and $\ell_2 = \frac{5}{\varepsilon}$.

## 4 Empirical Evaluation

In this section, along with Appendix F, we implement and empirically evaluate our randomized $(0.385 - \varepsilon)$-approximation algorithm (Alg. 2, FASTLS+GUIDEDRG) on two applications of size-constrained SM, and compare to several baselines in terms of objective value of solution and number of queries to $f$. In summary, our algorithm uses roughly twice the queries as the standard greedy algorithm, but obtains competitive objective values with an expensive local search that uses one to two orders of magnitude more queries. [1]

**Baselines.** 1) STANDARDGREEDY: the classical greedy algorithm [25], which often performs well empirically on non-monotone objectives despite having no theoretical guarantee. 2) RANDOM-GREEDY, the current state-of-the-art combinatorial algorithm as discussed extensively above. 3) The local search algorithm of Lee et al. [21], which is the only prior polynomial-time local search algorithm with a theoretical guarantee: ratio $1/4 - \varepsilon$ in $\mathcal{O}\left(k^5 \log(k)n/\varepsilon\right)$ queries. As our emphasis is on theoretical guarantees above $1/e$, we set $\varepsilon = 0.01$ for our algorithm, which yields ratio at least 0.375 in this evaluation. For Lee et al. [21], we set $\varepsilon = 0.1$, which is the standard value of the accuracy parameter in the literature – running their algorithm with $\varepsilon = 0.01$ produced identical results.

**Applications and datasets.** For instances of SM upon which to evaluate, we chose video summarization and maximum cut (MC). For video summarization, our objective is to select a subset of frames from a video to create a summary. As in Banihashem et al. [1] , we use a Determinantal Point Process objective function to select a diverse set of elements [20]. Maximum cut is a classical example of a non-monotone, submodular objective function. We run experiments on unweighted Erdős-Rényi (ER), Barabási-Albert (BA) and Watts-Strogatz (WS) graphs which have been used to model many real-world networks. The formal definition of problems, details of datasets, and hyperparameters of graph generation can be found in the Appendix F. In video summarization, there are $n = 100$ frames. On all the instances of maximum cut, the number of vertices $n = 10000$. The mean of 20 independent runs is plotted, and the shaded region represents one standard deviation about the mean.

**Results.** As shown in Figure 3 in this section, and Figure 5 and 6 in Appendix F, on both applications, FASTLS +GUIDEDRG produces solutions of higher objective value than STANDARDGREEDY, and also higher than RANDOMGREEDY. The objective values of FASTLS +GUIDEDRG often matches with Lee et al. [21] which performs the best; this agrees with the intuition that, empirically, local search is nearly optimal. In terms of queries, our algorithm uses roughly twice the number of queries

---

[1]Our code is available at https://gitlab.com/luciacyx/guided-rg.git.

as STANDARDGREEDY, but we improve on Lee et al. [21] typically by at least a factor of $10$ and often by more than a factor of $100$.

## 5 Discussion and Limitations

Prior to this work, the state-of-the-art combinatorial ratios were $1/e \approx 0.367$ and $0.283$ for size constrained and matroid constrained SM, respectively, both achieved by the RANDOMGREEDY algorithm. In this work, we show how to guide RANDOMGREEDY with a fast local search algorithm to achieve ratios $0.385$ and $0.305$, respectively, in $\mathcal{O}\left(kn/\varepsilon\right)$ queries. The resulting algorithm is practical and empirically outperforms both RANDOMGREEDY and standard greedy in objective value on several applications of SM. However, if $k$ is on the order of $n$, the query complexity is quadratic in $n$, which is too slow for modern data sizes. Therefore, an interesting question for future work is whether further improvements in the query complexity to achieve these ratios (or better ones) could be made.

In addition, we achieve the same approximation ratios and asymptotic query complexity with deterministic algorithms, achieved by guiding a different algorithm; moreover, we speed up the deterministic algorithm to $\mathcal{O}_{\varepsilon}(n \log k)$ by obtaining the guidance set in another way. This result is a partial answer to the limitation in the previous paragraph, as we achieve a ratio beyond $1/e$ in nearly linear query complexity. However, for all of our deterministic algorithms, there is an exponential dependence on $1/\varepsilon$, which makes these algorithms impractical and mostly of theoretical interest.

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

# A   Additional Preliminaries

## A.1   Constraints

In this paper, our focus lies on two constraints: size constraint and matroid constraint. For size constraint, we define the feasible subsets as $\mathcal{I}(k) = \{S \subseteq \mathcal{U} : |S| \leq k\}$. For matroid constraint, the definition is as follows:

**Definition A.1.** A matroid $\mathcal{M}$ is a pair $(\mathcal{U}, \mathcal{I})$, where $\mathcal{U}$ is the ground set and $\mathcal{I}$ is the independent sets with the following properties: (1) $\emptyset \in \mathcal{I}$; (2) hereditary property: $A \in \mathcal{I} \Rightarrow B \in \mathcal{I}, \forall B \subseteq A$; (3) exchange property: $A, B \in \mathcal{I}, |A| > |B| \Rightarrow \exists x \in A \setminus B, s.t. B + x \in \mathcal{I}$.

Specifically, we use $\mathcal{I}(\mathcal{M})$ to represent the independent sets of matroid $\mathcal{M}$. A maximal independent set in $\mathcal{I}(\mathcal{M})$ is called a basis. Let $k$ be the size of the maximal independent set.

In the following, we consider an extended matroid with $k$ dummy elements added to the ground set $\mathcal{M}' = (\mathcal{U}', \mathcal{I}')$. We show that SM on $\mathcal{M}'$ return the same solution as on $\mathcal{M}$.

**Lemma A.1.** Let $\mathcal{M} = (\mathcal{U}, \mathcal{I})$ be a matroid, and $\mathcal{U}' = \mathcal{U} \cup E$, where $E = \{e_0^1, \ldots, e_0^k\}$ and $e_0^i$ is a dummy element for each $i \in [k]$. Let $\mathcal{I}' = \bigcup_{S \in \mathcal{I}}\{S, S \cup \{e_0^1\}, \ldots, S \cup \{e_0^1, \ldots, e_0^{k-|S|}\}\}$. Then, $\mathcal{M}' = (\mathcal{U}', \mathcal{I}')$ is also a matroid and $\max_{S \in \mathcal{I}} f(S) = \max_{S \in \mathcal{I}'} f(S)$.

*Proof.* Firstly, we prove that $\mathcal{M}' = (\mathcal{U}', \mathcal{I}')$ is also a matroid by Definiton A.1.

Since $\emptyset \in \mathcal{I}$ and $\mathcal{I} \subseteq \mathcal{I}'$, it holds that $\emptyset \in \mathcal{I}'$.

Let $A' \in \mathcal{I}'$, and $A = A' \setminus E$. Then, $A \in \mathcal{I}$. For every $B' \subseteq A'$, since $(\mathcal{U}, \mathcal{I})$ is a matroid, $B = B' \setminus E \subseteq A \in \mathcal{I}$. Since $|B' \cap E| \leq |A' \cap E| \leq k - |A| \leq k - |B|$, it holds that $B \in \mathcal{I}'$ by the construction of $\mathcal{I}'$.

Let $A', B' \in \mathcal{I}'$ and $|A'| > |B'|$. Let $A = A' \setminus E$ and $B = B' \setminus E$. Then, $A, B \in \mathcal{I}$. If $|A| > |B|$, by Def. A.1, there exists $x \in A \setminus B$ s.t. $B + x \in \mathcal{I}$. Since $(B' + x) \setminus (B + x) \subseteq E$ and $|B'| < |A'| \leq k$, $B' + x \in \mathcal{I}'$. Otherwise, $|A| < |B|$, which indicates that $|A' \cap E| > |B' \cap E|$. Since $|B'| < |A'| \leq k$, by adding a dummy element $e_0 \in A' \cap E \setminus B'$ to $B'$, it holds that $B' + e_0 \in \mathcal{I}'$.

Thus, by Def. A.1, $\mathcal{M}' = (\mathcal{U}', \mathcal{I}')$ is also a matroid.

As for SM on $\mathcal{I}'$, since dummy element does not contribute to the objective value, it holds that, for every $S \in \mathcal{I}$, $f(S) = f(S \cup E')$, where $E' \subseteq E$. Then, $\{f(S) : S \in \mathcal{I}\} = \{f(S) : S \in \mathcal{I}'\}$. Further, $\max_{S \in \mathcal{I}} f(S) = \max_{S \in \mathcal{I}'} f(S)$. $\qquad\square$

## A.2   Technical Lemma

**Lemma A.2.** (Brualdi [4]) If $B_1$ and $B_2$ are finite bases, then there exists a bijection $\sigma : B_1 \setminus B_2 \to B_2 \setminus B_1$ such that $B_2 + e - \sigma(e)$ is a basis for all $e \in B_1 \setminus B_2$

**Lemma A.3.** Let $a \in \mathbb{R}^+$, $b, X_0 \in \mathbb{R}$. If $X_i \geq aX_{i-1} + b$ for every $i \in [k]$, then

$$X_k \geq a^k X_0 + \frac{b\left(1 - a^k\right)}{1 - a}.$$

*Proof.* By repeatedly implementing that $X_i \geq aX_{i-1} + b$, we can bound $X_k$ as follows,

$$
\begin{aligned}
X_k &\geq aX_{k-1} + b \\
&\geq a^2 X_{k-2} + ab + b \\
&\cdots \\
&\geq a^k X_0 + b\left(\sum_{i=0}^{k-1} a^i\right) \\
&= a^k X_0 + \frac{b\left(1 - a^k\right)}{1 - a}. \square
\end{aligned}
$$

**Lemma A.4.**

$$1 - \frac{1}{x} \leq \log(x) \leq x - 1, \qquad\qquad \forall x > 0$$

$$1 - \frac{1}{x+1} \geq e^{-\frac{1}{x}}, \qquad\qquad \forall x \in \mathbb{R}$$

$$(1-x)^{y-1} \geq e^{-xy}, \qquad\qquad \forall xy \leq 1$$

# B Query Complexity Analysis of the Continuous Algorithm in Buchbinder and Feldman [6]

---

**Algorithm 3:** Aided Measured Continuous Greedy $(f, P, Z, t_s)$ [6]

**input:** oracle $f$, a solvable down-closed polytope $P$, a set $Z \in \mathcal{U}$, $t_s \in (0,1)$

/\* Initialization \*/

1 Let $\bar{\delta}_1 \leftarrow t_s \cdot n^{-4}$ and $\bar{\delta}_2 \leftarrow (1-t_s) \cdot n^{-4}$
2 Let $t \leftarrow 0$ and $y(t) \leftarrow \mathbf{1}_\emptyset$

/\* Growing $y(t)$ \*/

3 **while** $t < 1$ **do**
4    **foreach** $u \in \mathcal{U}$ **do**
5       Let $w_u(t)$ be an estimate of $\mathbb{E}\left[\Delta(u|R(y(t)))\right]$ obtained by averaging the values of $\Delta(u|R(y(t)))$ for $r = \lceil 48n^6 \log(2n) \rceil$ independent samples of $R(y(t))$
6    **end**
7    Let $x(t) \leftarrow \begin{cases} \arg\max_{x \in P}\left\{\sum_{u \in \mathcal{U}\backslash Z} w_u(t) \cdot x_u(t) - \sum_{u \in Z} x_u(t)\right\} & \text{if } t \in [0, t_s), \\ \arg\max_{x \in P}\left\{\sum_{u \in \mathcal{U}} w_u(t) \cdot x_u(t)\right\} & \text{if } t \in [t_s, 1). \end{cases}$
8    Let $\delta_t$ be $\bar{\delta}_1$ when $t < t_s$ and $\bar{\delta}_2$ when $t \geq t_s$
9    Let $y(t + \delta_t) \leftarrow y(t) + \delta_t (\mathbf{1}_\mathcal{U} - y(t)) \circ x(t)$
10    Update $t \leftarrow t + \delta_t$
11 **end**
12 **return** $y(1)$

---

In this section, we analyze the query complexity of Aided Measured Continuous Greedy (Alg. 3) proposed by Buchbinder and Feldman [6], which is $\mathcal{O}\left(n^{11}\log(n)\right)$.

In Alg. 3, queries to the oracle $f$ only occur on Line 5. For each element $u$ in the ground set $\mathcal{U}$, $r = \lceil 48n^6 \log(2n) \rceil$ queries are made. These queries correspond to $r$ independent samples of $R(y(t))$ to estimate $\mathbb{E}\left[\Delta(u|R(y(t)))\right]$. Therefore, there are $nr = \mathcal{O}\left(n^7 \log(n)\right)$ queries for each iteration of the while loop (Line 3-11).

Time variable $t$ is increased by $\bar{\delta}_1 = t_s \cdot n^{-4}$, when $t < t_s$, and is increased by $\bar{\delta}_2 = (1-t_s) \cdot n^{-4}$, when $t \geq t_s$. Thus, there are a total of $2n^4$ iterations within the while loop. In conclusion, the total number of queries made by the algorithm is $\mathcal{O}\left(n^{11}\log(n)\right)$.

# C Analysis of Randomized Approximation Algorithm, Alg. 2

In this section, we provide a detailed analysis of our randomized approximation algorithm and its two components, FASTLS and GUIDEDRG. This section is organized as follows: Appendix C.1 analyzes the theoretical guarantee of single run of FASTLS (Appendix C.1.1), which is needed to show that it finds a good guidance set. Then, although it is not needed for our results, we show the FASTLS independently achieves an approximation ratio achieved for SMCC under monotone and non-monotone objectives (Appendix C.1.2).

In Appendix C.2, we provide pseudocode and formally prove the results for GUIDEDRG. Specifically, we solve the recurrences for both size and matroid constraint.

**Algorithm 4:** A fast local search algorithm with query complexity $\mathcal{O}\left(kn/\varepsilon\right)$.

---

1 **Procedure** FASTLS $(f, \mathcal{I}(\mathcal{M}), Z_0, \varepsilon)$**:**
2     **Input:** oracle $f$, matroid constraint $\mathcal{I}(\mathcal{M})$, an approximation result $Z_0$, accuracy parameter $\varepsilon$
3     **Initialize:** $Z \leftarrow Z_0$       `/* add dummy elements to Z until |Z| = k */`
4     **while** $\exists a \in Z, e \in \mathcal{U} \setminus Z, s.t. Z - a + e \in \mathcal{I}(\mathcal{M})$ *and* $\Delta(e|Z) - \Delta(a|Z \setminus a) \geq \frac{\varepsilon}{k} f\left(Z\right)$
5     **do**
6       $\big|$   $Z \leftarrow Z - a + e$
7     **end**
8     **return** $Z$

---

## C.1   Analysis of FASTLS (Alg. 4)

Pseudocode for FASTLS is provided in Alg. 4.

### C.1.1   Finding A Good Guidance Set – Proofs for Lemma 2.2 of Alg. 4 in Section 2.1

Recall that FASTLS (Alg. 4) takes a matroid constraint $\mathcal{I}(\mathcal{M})$ and an approximation result $Z_0$ as inputs, and outputs a local optimum $Z$. Here, we restate the theoretical guarantees of FASTLS (Lemma 2.2). Using the conclusions drawn from Lemma 2.2, we demonstrate in Corollary C.1 that $Z$ is a $((1+\varepsilon)\alpha, \alpha)$-guidance set. At the end of this section, we provide the proof for Lemma 2.2.

**Lemma 2.2.** Let $\varepsilon > 0$, and let $(f, \mathcal{I}(\mathcal{M}))$ be an instance of SM. The input set $Z_0$ is an $\alpha_0$-approximate solution to $(f, \mathcal{I}(\mathcal{M}))$. FASTLS (Alg. 4) returns a solution $Z$ with $\mathcal{O}\left(kn \log(1/\alpha_0)/\varepsilon\right)$ queries such that $f\left(S \cup Z\right) + f\left(S \cap Z\right) < (2+\varepsilon)f\left(Z\right)$, where $S \in \mathcal{I}(\mathcal{M})$.

**Corollary C.1.** Let $Z$ be the solution of FASTLS $(f, \mathcal{I}(\mathcal{M}), Z_0, \varepsilon)$. If $f\left(Z\right) < \alpha\text{OPT}$, $Z$ is a $((1+\varepsilon)\alpha, \alpha)$-guidance set.

*Proof of Corollary C.1.* By Lemma 2.2, let $S = O$ and $S = O \cap Z$ respectively, it holds that

$$f\left(O \cup Z\right) + f\left(O \cap Z\right) < (2+\varepsilon)f\left(Z\right),$$
$$f\left(O \cap Z\right) < (1+\varepsilon)f\left(Z\right).$$

If $f\left(Z\right) < \alpha\text{OPT}$, then

$$f\left(O \cup Z\right) + f\left(O \cap Z\right) < (2+\varepsilon)\alpha\text{OPT},$$
$$f\left(O \cap Z\right) < (1+\varepsilon)\alpha\text{OPT}.$$

By Definition 2.1, $Z$ is a $((1+\varepsilon)\alpha, \alpha)$-guidance set. $\qquad\square$

In the following, we prove Lemma 2.2 for Alg. 4.

*Proof of Lemma 2.2.* **Query Complexity.** For each successful replacement of elements on Line 6, it holds that $\Delta(e|Z) - \Delta(a|Z - a) \geq \frac{\varepsilon}{k} f\left(Z\right)$. By submodularity,

$$f\left(Z - a + e\right) - f\left(Z\right) = \Delta(e|Z - a) - \Delta(a|Z - a) \geq \Delta(e|Z) - \Delta(a|Z - a) \geq \frac{\varepsilon}{k} f\left(Z\right).$$

Hence, the oracle value $f(Z)$ is increased by a factor of at least $(1 + \varepsilon/k)$ after the swap. Therefore, there are at most $\left\lceil \log_{1+\varepsilon/k}\left(\frac{f(O)}{f(Z_0)}\right) \right\rceil$ iterations, since otherwise, it would entail $f\left(Z\right) > f\left(O\right)$, which contradicts the fact that $O$ is the optimal solution. Then, since the algorithm makes at most $\mathcal{O}\left(n\right)$ queries at each iteration, the query complexity can be bounded as follows,

$$\# \text{ queries} \leq \mathcal{O}\left(n \left\lceil \log_{1+\varepsilon/k}\left(\frac{f\left(O\right)}{f\left(Z_0\right)}\right) \right\rceil\right) \leq \mathcal{O}\left(n\frac{k}{\varepsilon} \log\left(\frac{1}{\alpha}\right)\right)$$
$$(f\left(Z_0\right) \geq \alpha f\left(O\right); \text{ Lemma A.4})$$

**Objective Value.** For any $S \in \mathcal{I}(\mathcal{M})$, we consider adding dummy elements into $S$ until $|S| = k$. By Lemma A.1, we consider $Z$ and $S$ are bases of matroid $\mathcal{M}$ with or without dummy elements.

Then, by Lemma A.2, there exits a bijection $\sigma : S \setminus Z \to Z \setminus S$ such that $Z + e - \sigma(e)$ is a basis for all $e \in S \setminus Z$. After the algorithm terminates, for every $e \in S \setminus Z$, it holds that, $\Delta(e|Z) - \Delta(\sigma(e)|Z - \sigma(e)) < \frac{\varepsilon}{k} f(Z)$. Then,

$$\varepsilon f(Z) > \sum_{e \in S \setminus Z} \Delta(e|Z) - \sum_{a \in Z \setminus S} \Delta(a|Z - a)$$

Let $\ell = |S \setminus Z| = |Z \setminus S|$, $S \setminus Z = \{e_1, \ldots, e_\ell\}$, $Z \setminus S = \{a_1, \ldots, a_\ell\}$. By submodularity,

$$\sum_{e \in S \setminus Z} \Delta(e|Z) = (f(Z + e_1) - f(Z)) + (f(Z + e_2) - f(Z)) + \ldots + (f(Z + e_\ell) - f(Z))$$

$$\geq (f(Z + e_1 + e_2) - f(Z)) + (f(Z + e_3) - f(Z)) + \ldots + (f(Z + e_\ell) - f(Z))$$

$$\geq \ldots$$

$$\geq f(Z + e_1 + \ldots + e_\ell) - f(Z) = f(S \cup Z) - f(Z). \text{ Also by submodularity,}$$

$$\sum_{a \in Z \setminus S} \Delta(a|Z - a) = \sum_{i=1}^{\ell} \Delta(a_i|Z - a_i) \leq \sum_{i=1}^{\ell} \Delta(a_i|Z - a_1 - \ldots - a_i) = f(Z) - f(S \cap Z)$$

Thus,

$$\varepsilon f(Z) > f(S \cup Z) - f(Z) - (f(Z) - f(S \cap Z))$$
$$\Rightarrow (2 + \varepsilon) f(Z) > f(S \cup Z) + f(S \cap Z).$$

$\square$

### C.1.2 Approximation Ratio achieved by FASTLS

In this section, we show that FASTLS can be employed independently to achieves approximation ratios of nearly $1/2$ and $1/4$ for both the monotone and non-monotone versions of the problem, respectively.

**Monotone submodular functions.** By employing FASTLS once, it returns an $\frac{1}{2+\varepsilon}$-approximation result in monotone cases.

**Theorem C.1.** Let $\varepsilon > 0$, and let $(f, \mathcal{I}(\mathcal{M}))$ be an instance of SM, where $f$ is monotone. The input set $Z_0$ is an $\alpha_0$-approximate solution to $(f, \mathcal{I}(\mathcal{M}))$. FASTLS (Alg. 4) returns a solution $Z$ such that $f(Z) \geq f(O)/(2 + \varepsilon)$ with $\mathcal{O}(kn \log(1/\alpha_0)/\varepsilon)$ queries.

*Proof.* By Lemma 2.2, set $S = O$, it holds that

$$(2 + \varepsilon) f(Z) > f(O \cup Z) + f(O \cap Z) \geq f(O),$$

where the last inequality follows by monotonicity and non-negativity. $\square$

**Non-monotone submodular functions.** For the non-monotone problem, 2 repetitions of FASTLS (Alg. 5) yields a ratio of $\frac{1}{4+2\varepsilon}$. The theoretical guarantees and the corresponding analysis are provided as follows. We remark that this is a primitive implementation of the guiding idea: the second run of FASTLS avoids the output of the first one.

---

**Algorithm 5:** An $1/(4 + 2\varepsilon)$-approximation algorithm with $\mathcal{O}(kn/\varepsilon)$

1 **Input:** oracle $f$, constraint $\mathcal{I}$, an approximation result $Z_0$, switch point $t$, error rate $\varepsilon$
2 $Z_1 \leftarrow$ FASTLS$(f, \mathcal{U}, \mathcal{I}, Z_0, \varepsilon)$      /* run FASTLS with ground set $\mathcal{U}$ */
3 $Z_2 \leftarrow$ FASTLS$(f, \mathcal{U} \setminus Z_1, \mathcal{I}, Z_0, \varepsilon)$    /* run FASTLS with ground set $\mathcal{U} \setminus Z_1$ */
4 **return** $Z \leftarrow \arg\max\{f(Z_1), f(Z_2)\}$

---

**Theorem C.2.** Let $\varepsilon > 0$, and let $(f, \mathcal{I}(\mathcal{M}))$ be an instance of SM, where $f$ is not necessarily monotone. The input set $Z_0$ is an $\alpha_0$-approximate solution to $(f, \mathcal{I}(\mathcal{M}))$. Alg. 5 returns a solution $Z$ such that $f(Z) \geq f(O)/(4 + 2\varepsilon)$ with $\mathcal{O}(kn \log(1/\alpha_0)/\varepsilon)$ queries.

*Proof.* By repeated application of Lemma 2.2 for the two calls of FASTLS in Alg. 5, it holds that

$$f(O \cup Z_1) + f(O \cap Z_1) < (2 + \varepsilon)f(Z_1)$$
$$f((O \setminus Z_1) \cup Z_2) + f((O \setminus Z_1) \cap Z_2) < (2 + \varepsilon)f(Z_2)$$

By summing up the above two inequalities, it holds that

$$
\begin{aligned}
(4 + \varepsilon)f(Z) &\geq (2 + \varepsilon)f(Z_1) + (2 + \varepsilon)f(Z_2) \\
&\geq f(O \cup Z_1) + f(O \cap Z_1) + f((O \setminus Z_1) \cup Z_2) + f((O \setminus Z_1) \cap Z_2) \\
&\geq f(O \cup Z_1) + f((O \setminus Z_1) \cup Z_2) + f(O \cap Z_1) & \text{(nonnegativity)} \\
&\geq f(O \setminus Z_1) + f(O \cap Z_1) & \text{(submodularity)} \\
&\geq f(O). & \text{(submodularity)}
\end{aligned}
$$

$\square$

## C.2 Pseudocode of GUIDEDRG (Alg. 6) and its Analysis

---

**Algorithm 6:** An algorithm guided by an $(\alpha, \beta)$-guidance set $Z$ with $\mathcal{O}(kn)$ queries

---
1 **Procedure** GUIDEDRG $(f, \mathcal{I}, Z, t)$**:**
2     **Input:** oracle $f$, constraint $\mathcal{I}$, guidance set $Z$, switch point $t$
3     $A_0 \leftarrow k$ dummy elements         /* Equivalent to an empty set */
4     **for** $i \leftarrow 1$ *to* $k$ **do**
5         **if** $i \leq t \cdot k$ **then** $M_i \leftarrow \arg\max_{M \subseteq \mathcal{U} \setminus (A_{i-1} \cup Z), M \text{ is a basis}} \sum_{a \in M} \Delta(a | A_{i-1})$
6         **else** $M_i \leftarrow \arg\max_{M \subseteq \mathcal{U} \setminus A_{i-1}, M \text{ is a basis}} \sum_{a \in M} \Delta(a | A_{i-1})$
7         **if** $\mathcal{I}$ *represents the size constraint* **then**
8             $a_i \leftarrow$ randomly pick an element from $M_i$
9             $A_i \leftarrow A_{i-1} + a_i - e_0$         /* $e_0$ is the dummy element */
10         **else if** $\mathcal{I}$ *represents the matroid constraint* **then**
11             $\sigma_i \leftarrow$ a bijection from $M_i$ to $A_{i-1}$, where $A_{i-1} + x - \sigma_i(x) \in \mathcal{I}(\mathcal{M}), \forall x \in M_i$
12             $e_i \leftarrow$ randomly pick an element from $M_i$
13             $A_i \leftarrow A_{i-1} + e_i - \sigma_i(e_i)$
14     **end**
15     **return** $A_k$

---

In this section, we present the pseudocode for GUIDEDRG as Alg. 6. Then, we provide the detailed proof of Lemma 2.3 in Appendix C.2.1, which addresses size constraints. Finally, we analyze the algorithm under matroid constraints and provide the guarantees and its analysis in Appendix C.2.2.

### C.2.1 GUIDEDRG under Size Constraints

In Sec. 2.2, we introduce the intuition behind GUIDEDRG under size constraint. Below, we reiterate theoretical guarantees achieved by GUIDEDRG under size constraints and provide the detailed analysis.

**Lemma 2.3.** With an input size constraint $\mathcal{I}$ and a $((1 + \varepsilon)\alpha, \alpha)$-guidance set $Z$, GUIDEDRG returns set $A_k$ with $\mathcal{O}(kn)$ queries, *s.t.* $\mathbb{E}[f(A_k)] \geq \left[ \left(2 - t - \frac{1}{k}\right) \left(1 - \frac{1}{k}\right) e^{t-1} - e^{-1} - (1 + \varepsilon)\alpha \left( \left(1 - \frac{1}{k}\right)^2 e^{t-1} - e^{-1} \right) - \alpha \left( \left(1 + \frac{1-t}{1 - \frac{1}{k}}\right) e^{t-1} - \left(2 - \frac{1}{k}\right) e^{-1} \right) \right] f(O).$

We provide the recurrence of $f((O \setminus Z) \cup A_i)$, $f(O \cup A_i)$ and $f(A_i)$ in Lemmata C.3 and C.4 and their analysis below to help prove Lemma 2.3 under size constraint.

**Lemma C.3.** When $0 < i \leq t \cdot k$, it holds that

$$\mathbb{E}[f((O \setminus Z) \cup A_i)] \geq \left(1 - \frac{1}{k}\right) \mathbb{E}[f((O \setminus Z) \cup A_{i-1})] + \frac{1}{k}[f(O \setminus Z) - f(O \cup Z)],$$

$$\mathbb{E}[f(O \cup A_i)] \geq \left(1 - \frac{1}{k}\right) \mathbb{E}[f(O \cup A_{i-1})] + \frac{1}{k}[f(O) - f(O \cup Z)].$$

When $t \cdot k < i \leq k$, it holds that

$$\mathbb{E}\left[f\left(O \cup A_i\right)\right] \geq \left(1 - \frac{1}{k}\right) \mathbb{E}\left[f\left(O \cup A_{i-1}\right)\right]$$

*Proof.* At iteration $i$, condition on a given $A_{i-1}$. When $i \leq tk$, $A_{i-1} \cap Z = \emptyset$ and $M_i$ is selected out of $A_{i-1} \cup Z$, so

$$(O \cup Z) \cap ((O \setminus Z) \cup A_{i-1} \cup M_i) = O \setminus Z \tag{3}$$

$$((O \cup Z) \cap (O \cup A_{i-1} \cup M_i) = O. \tag{4}$$

Then,

$$\mathbb{E}\left[f\left((O \setminus Z) \cup A_i\right) \mid A_{i-1}\right] = \frac{1}{k} \sum_{x \in M_i} f\left((O \setminus Z) \cup A_{i-1} \cup \{x\}\right) \quad \text{(selection of next element)}$$

$$\geq \frac{1}{k}\left[(k-1)f\left((O \setminus Z) \cup A_{i-1}\right) + f\left((O \setminus Z) \cup A_{i-1} \cup M_i\right)\right] \quad \text{(submodularity)}$$

$$\geq \frac{1}{k}\left[(k-1)f\left((O \setminus Z) \cup A_{i-1}\right) + f\left(O \setminus Z\right) + f\left(O \cup Z \cup A_{i-1} \cup M_i\right) - f\left(O \cup Z\right)\right]$$
$$\text{(submodularity)}$$

$$\geq \frac{1}{k}\left[(k-1)f\left((O \setminus Z) \cup A_{i-1}\right) + f\left(O \setminus Z\right) - f\left(O \cup Z\right)\right] \quad \text{(nonnegativity)}$$

$$\mathbb{E}\left[f\left(O \cup A_i\right) \mid A_{i-1}\right] = \frac{1}{k} \sum_{x \in M_i} f\left(O \cup A_{i-1} \cup \{x\}\right)$$

$$\geq \frac{1}{k}\left[(k-1)f\left(O \cup A_{i-1}\right) + f\left(O \cup A_{i-1} \cup M_i\right)\right] \quad \text{(submodularity)}$$

$$\geq \frac{1}{k}\left[(k-1)f\left(O \cup A_{i-1}\right) + f\left(O\right) + f\left(O \cup Z \cup A_{i-1} \cup M_i\right) - f\left(O \cup Z\right)\right] \quad \text{(submodularity)}$$

$$\geq \frac{1}{k}\left[(k-1)f\left(O \cup A_{i-1}\right) + f\left(O\right) - f\left(O \cup Z\right)\right] \quad \text{(nonnegativity)}$$

When $i > tk$, it holds that

$$\mathbb{E}\left[f\left(O \cup A_i\right) \mid A_{i-1}\right] = \frac{1}{k} \sum_{x \in M_i} f\left(O \cup A_{i-1} \cup \{x\}\right)$$

$$\geq \frac{1}{k}\left[(k-1)f\left(O \cup A_{i-1}\right) + f\left(O \cup A_{i-1} \cup M_i\right)\right] \quad \text{(submodularity)}$$

$$\geq \left(1 - \frac{1}{k}\right)f\left(O \cup A_{i-1}\right) \quad \text{(nonnegativity)}$$

By unconditioning $A_{i-1}$, the lemma is proved. $\qquad \square$

**Lemma C.4.** When $0 < i \leq t \cdot k$, it holds that

$$\mathbb{E}\left[f\left(A_i\right)\right] - \mathbb{E}\left[f\left(A_{i-1}\right)\right] \geq \frac{1}{k}\left(\mathbb{E}\left[f\left((O \setminus Z) \cup A_{i-1}\right)\right] - \mathbb{E}\left[f\left(A_{i-1}\right)\right]\right).$$

When $t \cdot k < i \leq k$, it holds that

$$\mathbb{E}\left[f\left(A_i\right)\right] - \mathbb{E}\left[f\left(A_{i-1}\right)\right] \geq \frac{1}{k}\left(\mathbb{E}\left[f\left(O \cup A_{i-1}\right)\right] - \mathbb{E}\left[f\left(A_{i-1}\right)\right]\right).$$

*Proof.* Given $A_{i-1}$ at iteration $i$. When $i \leq t \cdot k$, it holds that

$$\mathbb{E}\left[f\left(A_i\right) - f\left(A_{i-1}\right) \mid A_{i-1}\right] = \frac{1}{k} \sum_{x \in M_i} \Delta(x|A_{i-1})$$

$$\geq \frac{1}{k} \sum_{x \in O \setminus (A_{i-1} \cup Z)} \Delta(x|A_{i-1}) \quad \text{(Line 5 in Alg. 6)}$$

$$\geq \frac{1}{k}\left[f\left((O \setminus Z) \cup A_{i-1}\right) - f\left(A_{i-1}\right)\right]. \quad \text{(submodularity)}$$

When $i > t \cdot k$, it holds that

$$\mathbb{E}\left[f\left(A_i\right) - f\left(A_{i-1}\right) \mid A_{i-1}\right] = \frac{1}{k}\sum_{x \in M_i} \Delta(x|A_{i-1})$$

$$\geq \frac{1}{k}\sum_{x \in O \setminus A_{i-1}} \Delta(x|A_{i-1}) \qquad\qquad \text{(Line 6 in Alg. 6)}$$

$$\geq \frac{1}{k}\left[f\left(O \cup A_{i-1}\right) - f\left(A_{i-1}\right)\right]. \qquad\qquad \text{(submodularity)}$$

By unconditioning $A_{i-1}$, the lemma is proved. $\qquad\square$

*Proof of Lemma 2.3.* It follows from Lemma C.3 and the closed form for a recurrence provided in Lemma A.3 that

$$
\begin{cases}
\mathbb{E}\left[f\left((O \setminus Z) \cup A_i\right)\right] \geq f\left(O \setminus Z\right) - \left(1 - \left(1 - \frac{1}{k}\right)^i\right)f\left(O \cup Z\right), & \forall 0 < i \leq tk \\[2mm]
\mathbb{E}\left[f\left(O \cup A_i\right)\right] \geq \left(1 - \frac{1}{k}\right)^{i - \lfloor tk \rfloor}\left[f\left(O\right) - \left(1 - \left(1 - \frac{1}{k}\right)^{\lfloor tk \rfloor}\right)f\left(O \cup Z\right)\right], & \forall tk < i \leq k
\end{cases}
$$

$$(5)$$

With the above inequalities, we can solve the recursion in Lemma C.4 as follows,

$$\mathbb{E}\left[f\left(A_{\lfloor tk \rfloor}\right)\right] \overset{(a)}{\geq} \left(1 - \left(1 - \frac{1}{k}\right)^{\lfloor tk \rfloor}\right)f\left(O \setminus Z\right) - \left(1 - \left(1 - \frac{1}{k}\right)^{\lfloor tk \rfloor} - t\left(1 - \frac{1}{k}\right)^{\lfloor tk \rfloor - 1}\right)f\left(O \cup Z\right)$$

$$\mathbb{E}\left[f\left(A_k\right)\right] \overset{(b)}{\geq} \left(1 - \frac{1}{k}\right)^{k - \lfloor tk \rfloor}\mathbb{E}\left[f\left(A_{\lfloor tk \rfloor}\right)\right] + (1 - t)\left(1 - \frac{1}{k}\right)^{k - \lfloor tk \rfloor - 1}\left[f\left(O\right) - \left(1 - \left(1 - \frac{1}{k}\right)^{\lfloor tk \rfloor}\right)f\left(O \cup Z\right)\right]$$

$$\geq (1 - t)\left(1 - \frac{1}{k}\right)^{k - \lfloor tk \rfloor - 1}f\left(O\right) + \left(\left(1 - \frac{1}{k}\right)^{k - \lfloor tk \rfloor} - \left(1 - \frac{1}{k}\right)^{k}\right)f\left(O \setminus Z\right)$$

$$- \left(\left(1 + \frac{1 - t}{1 - \frac{1}{k}}\right)\left(1 - \frac{1}{k}\right)^{k - \lfloor tk \rfloor} - \left(2 - \frac{1}{k}\right)\left(1 - \frac{1}{k}\right)^{k - 1}\right)f\left(O \cup Z\right)$$

$$\text{(Inequality } (a))$$

$$\geq (1 - t)\left(1 - \frac{1}{k}\right)^{(1 - t)k}f\left(O\right) + \left(\left(1 - \frac{1}{k}\right)^{(1 - t)k + 1} - \left(1 - \frac{1}{k}\right)^{k}\right)f\left(O \setminus Z\right)$$

$$- \left(\left(1 + \frac{1 - t}{1 - \frac{1}{k}}\right)\left(1 - \frac{1}{k}\right)^{(1 - t)k} - \left(2 - \frac{1}{k}\right)\left(1 - \frac{1}{k}\right)^{k - 1}\right)f\left(O \cup Z\right)$$

$$(tk - 1 < \lfloor tk \rfloor \leq tk)$$

$$\geq (1 - t)\left(1 - \frac{1}{k}\right)e^{t - 1}f\left(O\right) + \left(\left(1 - \frac{1}{k}\right)^2 e^{t - 1} - e^{-1}\right)f\left(O \setminus Z\right)$$

$$- \left(\left(1 + \frac{1 - t}{1 - \frac{1}{k}}\right)e^{t - 1} - \left(2 - \frac{1}{k}\right)e^{-1}\right)f\left(O \cup Z\right) \qquad \text{(nonnegativity;Lemma A.4)}$$

$$\geq (1 - t)\left(1 - \frac{1}{k}\right)e^{t - 1}f\left(O\right) + \left(\left(1 - \frac{1}{k}\right)^2 e^{t - 1} - e^{-1}\right)\left(f\left(O\right) - f\left(O \cap Z\right)\right)$$

$$- \left(\left(1 + \frac{1 - t}{1 - \frac{1}{k}}\right)e^{t - 1} - \left(2 - \frac{1}{k}\right)e^{-1}\right)f\left(O \cup Z\right) \qquad \text{(submodularity)}$$

$$= \left(\left(2 - t - \frac{1}{k}\right)\left(1 - \frac{1}{k}\right)e^{t - 1} - e^{-1}\right)f\left(O\right) - \left(\left(1 - \frac{1}{k}\right)^2 e^{t - 1} - e^{-1}\right)f\left(O \cap Z\right)$$

$$- \left(\left(1 + \frac{1 - t}{1 - \frac{1}{k}}\right)e^{t - 1} - \left(2 - \frac{1}{k}\right)e^{-1}\right)f\left(O \cup Z\right),$$

where Inequality $(a)$ and $(b)$ follow from Inequality 5, Lemma C.4 and A.3. $\qquad\square$

### C.2.2 GUIDEDRG under Matroid Constraints

---
**Algorithm 7:** RANDOMGREEDY for Matroid
---
1 **Procedure** RANDOMGREEDY $(f, \mathcal{M})$**:**
2     **Input:** oracle $f$, matroid constraint $\mathcal{M}$
3     **Initialize:** $A_0 \leftarrow$ arbitrary basis in $\mathcal{I}(\mathcal{M})$
4     **for** $i \leftarrow 1$ *to* $k$ **do**
5         $M_i \leftarrow \arg\max_{S \subseteq \mathcal{U}, S \text{ is a basis}} \sum_{x \in S} \Delta(x|A_{i-1})$
6         $\sigma \leftarrow$ a bijection from $M_i$ to $A_{i-1}$
7         $x_i \leftarrow$ a uniformly random element from $M_i$
8         $A_i \leftarrow A_{i-1} + x_i - \sigma(x_i)$
9     **end**
10     **return** $A_k$
---

**Discussion about Intuition behind GUIDEDRG under Matroid Constraints.** The pseudocode for RANDOMGREEDY under matroid constraints is provided in Alg. 7. To deal with the feasibility for matroid constraints, Alg. 7 starts with an arbitrary basis and builds the solution by randomly swapping the elements in ground set with a candidate basis. The analysis of it proceeds according to two main recurrences.

$$1) \ \mathbb{E}\left[f\left(A_i\right) - f\left(A_{i-1}\right)\right] \geq \frac{1}{k}\mathbb{E}\left[f\left(O \cup A_{i-1}\right) - 2f\left(A_{i-1}\right)\right],$$

$$2) \ \mathbb{E}\left[f\left(O \cup A_i\right)\right] \geq \left(1 - \frac{2}{k}\right)\mathbb{E}\left[f\left(O \cup A_{i-1}\right)\right] + \frac{1}{k}\mathbb{E}\left[f\left(O\right) + f\left(O \cup A_{i-1} \cup M_i\right)\right].$$

Fig. 4(a) depicts the worse-case behavior of $\mathbb{E}\left[f\left(A_i\right)\right]$ and $\mathbb{E}\left[f\left(O \cup A_i\right)\right]$. As discussed in Section 2.2, we consider improving the degradation of $\mathbb{E}\left[f\left(O \cup A_i\right)\right]$ by selecting elements from outside of an $(\alpha + \varepsilon, \alpha)$-guidance set $Z$ to enhance the lower bound of $\mathbb{E}\left[f\left(O \cup A_{i-1} \cup M_i\right)\right]$. The blue line in Fig. 4(b) illustrated the improvement of $\mathbb{E}\left[f\left(O \cup A_i\right)\right]$ with an $(\alpha + \varepsilon, \alpha)$-guidance set. However, restricting the selection only to elements outside $Z$ restricts the increase in $\mathbb{E}\left[f\left(A_i\right)\right]$ to the difference between $\mathbb{E}\left[f\left((O \setminus Z) \cup A_{i-1}\right)\right]$ and $\mathbb{E}\left[f\left(A_{i-1}\right)\right]$. This restriction is illustrated by the red line in Figure 4(b), indicating degradation in $\mathbb{E}\left[f\left((O \setminus Z) \cup A_i\right)\right]$.

To benefit from the improved degradation of $\mathbb{E}\left[f\left(O \cup A_i\right)\right]$, we consider transitioning to selecting elements from the whole ground set at a suitable point. The blue line in Fig. 4(b) illustrates how $\mathbb{E}\left[f\left(O \cup A_i\right)\right]$ degrades before and after we switch, and the orange line illustrates the evolution of $\mathbb{E}\left[f\left(A_i\right)\right]$. Even starting with an inferior selection at the first stage, we still get an overall improvement on the objective value.

We provide the updated recursions for $f\left((O \setminus Z) \cup A_i\right)$, $f\left(O \cup A_i\right)$ and $f\left(A_i\right)$ in Lemma C.5 and C.6 below. Then, the closed form of the solution value, derived from these lemmata, is presented in Lemma C.7. After that, we prove the approximation ratio of the randomized algorithm under matroid constraint.

**Lemma C.5.** When $0 < i \leq t \cdot k$, it holds that

$$\mathbb{E}\left[f\left((O \setminus Z) \cup A_i\right)\right] \geq \left(1 - \frac{2}{k}\right)\mathbb{E}\left[f\left((O \setminus Z) \cup A_{i-1}\right)\right] + \frac{1}{k}\left[2f\left(O \setminus Z\right) - f\left(O \cup Z\right)\right],$$

$$\mathbb{E}\left[f\left(O \cup A_i\right)\right] \geq \left(1 - \frac{2}{k}\right)\mathbb{E}\left[f\left(O \cup A_{i-1}\right)\right] + \frac{1}{k}\left[2f\left(O\right) - f\left(O \cup Z\right)\right].$$

When $t \cdot k < i \leq k$, it holds that

$$\mathbb{E}\left[f\left(O \cup A_i\right)\right] \geq \left(1 - \frac{2}{k}\right)\mathbb{E}\left[f\left(O \cup A_{i-1}\right)\right] + \frac{1}{k}f\left(O\right)$$

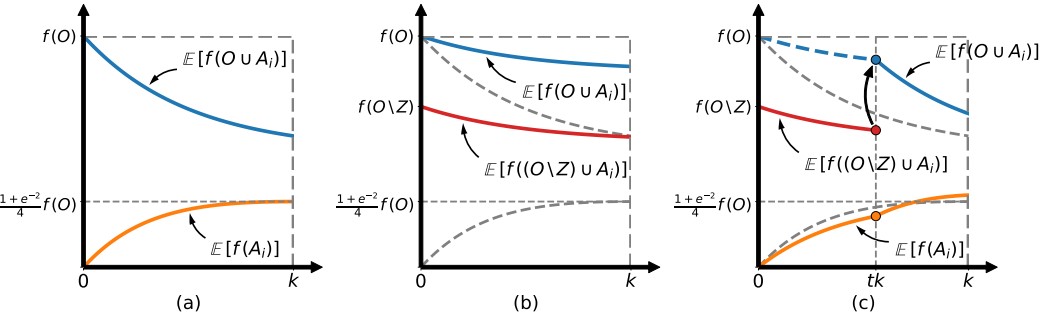

Figure 4: This set of figures indicates how guiding benefits RANDOMGREEDY under matroid constraints. The figure (a) depicts the evolution of $f\left(O \cup A_i\right)$ and $f\left(A_i\right)$ with RANDOMGREEDY. The figure (b) illustrates how the degradation of $f\left(O \cup A_i\right)$ changes as we introduce an $(0.305 + \varepsilon, 0.305)$-guidance set. Additionally, we also need to consider the degradation of $f\left((O \setminus Z) \cup A_i\right)$, which is the value that the solution approaches with the guidance. The figure (c) shows the updated degradation with a switch point $tk$, where the algorithm starts with guidance and then switches to running without guidance. It demonstrates that even though the value of $A_i$ decreases initially when the selection starts outside of $Z$, it benefits from the improved degradation of $f\left(O \cup A_i\right)$ upon switching back to the original algorithm.

*Proof.* When $i \leq tk$, it holds that

$$\mathbb{E}\left[f\left((O \setminus Z) \cup A_i\right) \mid A_{i-1}\right] = \frac{1}{k} \sum_{x \in M_i} f\left((O \setminus Z) \cup (A_{i-1} + x - \sigma_i(x))\right)$$

$$\geq \frac{1}{k} \sum_{x \in M_i} \left[\Delta(x|(O \setminus Z) \cup A_{i-1}) + f\left((O \setminus Z) \cup (A_{i-1} - \sigma_i(x))\right)\right] \qquad \text{(submodularity)}$$

$$\geq \frac{1}{k}\left[f\left((O \setminus Z) \cup A_{i-1} \cup M_i\right) - f\left((O \setminus Z) \cup A_{i-1}\right) + (k-1)f\left((O \setminus Z) \cup A_{i-1}\right) + f\left(O \setminus Z\right)\right]$$
$$\text{(submodularity)}$$

$$\geq \frac{1}{k}\left[(k-2)f\left((O \setminus Z) \cup A_{i-1}\right) + 2f\left(O \setminus Z\right) - f\left(O \cup Z\right)\right] \qquad \text{(submodularity; nonnegativity)}$$

$$\mathbb{E}\left[f\left(O \cup A_i\right) \mid A_{i-1}\right] = \frac{1}{k} \sum_{x \in M_i} f\left(O \cup (A_{i-1} + x - \sigma_i(x))\right)$$

$$\geq \frac{1}{k} \sum_{x \in M_i} \left[\Delta(x|O \cup A_{i-1}) + f\left(O \cup (A_{i-1} - \sigma_i(x))\right)\right] \qquad \text{(submodularity)}$$

$$\geq \frac{1}{k}\left[f\left(O \cup A_{i-1} \cup M_i\right) - f\left(O \cup A_{i-1}\right) + (k-1)f\left(O \cup A_{i-1}\right) + f\left(O\right)\right] \qquad \text{(submodularity)}$$

$$\geq \frac{1}{k}\left[(k-2)f\left(O \cup A_{i-1}\right) + 2f\left(O\right) - f\left(O \cup Z\right)\right] \qquad \text{(submodularity; nonnegativity)}$$

When $i > tk$, it holds that

$$\mathbb{E}\left[f\left(O \cup A_i\right) \mid A_{i-1}\right] = \frac{1}{k} \sum_{x \in M_i} f\left(O \cup (A_{i-1} + x - \sigma_i(x))\right)$$

$$\geq \frac{1}{k} \sum_{x \in M_i} \left[\Delta(x|O \cup A_{i-1}) + f\left(O \cup (A_{i-1} - \sigma_i(x))\right)\right] \qquad \text{(submodularity)}$$

$$\geq \frac{1}{k}\left[f\left(O \cup A_{i-1} \cup M_i\right) - f\left(O \cup A_{i-1}\right) + (k-1)f\left(O \cup A_{i-1}\right) + f\left(O\right)\right] \qquad \text{(submodularity)}$$

$$\geq \frac{1}{k}\left[(k-2)f\left(O \cup A_{i-1}\right) + f\left(O\right)\right] \qquad \text{(nonnegativity)}$$

By unconditioning $A_{i-1}$, the lemma is proved. $\qquad \square$

**Lemma C.6.** When $0 < i \leq t \cdot k$, it holds that

$$\mathbb{E}\left[f\left(A_i\right)\right] - \mathbb{E}\left[f\left(A_{i-1}\right)\right] \geq \frac{1}{k}\left(\mathbb{E}\left[f\left(\left(O \setminus Z\right) \cup A_{i-1}\right)\right] - 2\mathbb{E}\left[f\left(A_{i-1}\right)\right]\right).$$

When $t \cdot k < i \leq k$, it holds that

$$\mathbb{E}\left[f\left(A_i\right)\right] - \mathbb{E}\left[f\left(A_{i-1}\right)\right] \geq \frac{1}{k}\left(\mathbb{E}\left[f\left(O \cup A_{i-1}\right)\right] - 2\mathbb{E}\left[f\left(A_{i-1}\right)\right]\right).$$

*Proof.* Given $A_{i-1}$ at iteration $i$. When $i \leq t \cdot k$, since $O$ is a base, $O \setminus (A_{i-1} \cup Z)$ with dummy elements is also a base. It holds that

$$\mathbb{E}\left[f\left(A_i\right) - f\left(A_{i-1}\right) \mid A_{i-1}\right] = \frac{1}{k}\sum_{x \in M_i}\left[f\left(A_{i-1} + x - \sigma_i(x)\right) - f\left(A_{i-1}\right)\right]$$

$$\geq \frac{1}{k}\sum_{x \in M_i}\left[\Delta(x|A_{i-1}) + f\left(A_{i-1} - \sigma_i(x)\right) - f\left(A_{i-1}\right)\right] \quad \text{(submodularity)}$$

$$\geq \frac{1}{k}\sum_{x \in O \setminus (A_{i-1} \cup Z)}\Delta(x|A_{i-1}) + \frac{1}{k}\sum_{x \in M_i}\left[f\left(A_{i-1} - \sigma_i(x)\right) - f\left(A_{i-1}\right)\right] \quad \text{(Line 5 in Alg. 6)}$$

$$\geq \frac{1}{k}\left[f\left(\left(O \setminus Z\right) \cup A_{i-1}\right) - f\left(A_{i-1}\right)\right] - \frac{1}{k}f\left(A_{i-1}\right). \quad \text{(submodularity)}$$

When $i > t \cdot k$, it holds that

$$\mathbb{E}\left[f\left(A_i\right) - f\left(A_{i-1}\right) \mid A_{i-1}\right] = \frac{1}{k}\sum_{x \in M_i}\left[f\left(A_{i-1} + x - \sigma_i(x)\right) - f\left(A_{i-1}\right)\right]$$

$$\geq \frac{1}{k}\sum_{x \in M_i}\left[\Delta(x|A_{i-1}) + f\left(A_{i-1} - \sigma_i(x)\right) - f\left(A_{i-1}\right)\right] \quad \text{(submodularity)}$$

$$\geq \frac{1}{k}\sum_{x \in O}\Delta(x|A_{i-1}) + \frac{1}{k}\sum_{x \in M_i}\left[f\left(A_{i-1} - \sigma_i(x)\right) - f\left(A_{i-1}\right)\right] \quad \text{(Line 6 in Alg. 6)}$$

$$\geq \frac{1}{k}\left[f\left(O \cup A_{i-1}\right) - f\left(A_{i-1}\right)\right] - \frac{1}{k}f\left(A_{i-1}\right). \quad \text{(submodularity)}$$

$\square$

**Lemma C.7.** With an input matroid constraint $\mathcal{I}$ and a $((1 + \varepsilon)\alpha, \alpha)$-guidance set $Z$, GUIDEDRG returns set $A_k$ with $\mathcal{O}(kn)$ queries, s.t. $\mathbb{E}\left[f\left(A_k\right)\right] \geq \frac{1}{2}\left(\frac{1}{2} + \left(\frac{3}{2} - t - \frac{1}{k}\right)\left(1 - \frac{2}{k}\right)e^{2(t-1)} - e^{-2} - (1 + \varepsilon)\alpha\left(\left(1 - \frac{2}{k}\right)^2 e^{2(t-1)} - e^{-2}\right)\right.$
$\left.-\alpha\left(\left(\frac{1}{2} + \frac{1-t}{1-\frac{2}{k}}\right)e^{2(t-1)} - \left(\frac{3}{2} - \frac{1}{k}\right)e^{-2}\right)\right)f\left(O\right).$

*Proof.* It follows from Lemma C.5 and the closed form for a recurrence provided in Lemma A.3 that

$$\begin{cases}
\mathbb{E}\left[f\left(\left(O \setminus Z\right) \cup A_i\right)\right] \geq f\left(O \setminus Z\right) - \frac{1}{2}\left(1 - \left(1 - \frac{2}{k}\right)^i\right)f\left(O \cup Z\right), & \forall 0 < i \leq tk \\[2mm]
\mathbb{E}\left[f\left(O \cup A_i\right)\right] \geq \frac{1}{2}\left(1 + \left(1 - \frac{2}{k}\right)^{i - \lfloor tk \rfloor}\right)f\left(O\right) - \frac{1}{2}\left(\left(1 - \frac{2}{k}\right)^{i - \lfloor tk \rfloor} - \left(1 - \frac{2}{k}\right)^i\right)f\left(O \cup Z\right), & \forall tk < i \leq k
\end{cases}$$
$$(6)$$

Then, by solving the recursion in Lemma C.6 with the above inequalities, it holds that

$$\mathbb{E}\left[f\left(A_{\lfloor tk \rfloor}\right)\right] \overset{(a)}{\geq} \frac{1}{2}\left[1 - \left(1 - \frac{2}{k}\right)^{\lfloor tk \rfloor}\right]f\left(O \setminus Z\right) - \left[\frac{1}{4} - \frac{1}{4}\left(1 - \frac{2}{k}\right)^{\lfloor tk \rfloor} - \frac{t}{2}\left(1 - \frac{2}{k}\right)^{\lfloor tk \rfloor - 1}\right]f\left(O \cup Z\right)$$

$$\mathbb{E}\left[f\left(A_k\right)\right] \overset{(b)}{\geq} \left(1 - \frac{2}{k}\right)^{k - \lfloor tk \rfloor}\mathbb{E}\left[f\left(A_{\lfloor tk \rfloor}\right)\right] + \frac{1}{2}\left[\frac{1}{2} + \left(\frac{1}{2} - t + \frac{1}{k}\right)\left(1 - \frac{2}{k}\right)^{k - \lfloor tk \rfloor - 1}\right]f(O)$$

$$-\frac{1-t}{2}\left[\left(1-\frac{2}{k}\right)^{k-\lfloor tk\rfloor-1}-\left(1-\frac{2}{k}\right)^{k-1}\right]f(O\cup Z)$$

$$\geq\frac{1}{2}\left[\left(\left(1-\frac{2}{k}\right)^{k-\lfloor tk\rfloor}-\left(1-\frac{2}{k}\right)^{k}\right)f(O\setminus Z)+\left(\frac{1}{2}+\left(\frac{1}{2}-t+\frac{1}{k}\right)\left(1-\frac{2}{k}\right)^{k-\lfloor tk\rfloor-1}\right)f(O)\right.$$

$$\left.-\left(\left(\frac{1}{2}+\frac{1-t}{1-\frac{2}{k}}\right)\left(1-\frac{2}{k}\right)^{k-\lfloor tk\rfloor}-\left(\frac{3}{2}-\frac{1}{k}\right)\left(1-\frac{2}{k}\right)^{k-1}\right)\right]f(O\cup Z)$$

$$\text{(Inequality }(a)\text{)}$$

$$\geq\frac{1}{2}\left[\left(\left(1-\frac{2}{k}\right)^{(1-t)k+1}-\left(1-\frac{2}{k}\right)^{k}\right)f(O\setminus Z)+\left(\frac{1}{2}+\left(\frac{1}{2}-t+\frac{1}{k}\right)\left(1-\frac{2}{k}\right)^{(1-t)k}\right)f(O)\right.$$

$$\left.-\left(\left(\frac{1}{2}+\frac{1-t}{1-\frac{2}{k}}\right)\left(1-\frac{2}{k}\right)^{(1-t)k}-\left(\frac{3}{2}-\frac{1}{k}\right)\left(1-\frac{2}{k}\right)^{k-1}\right)\right]f(O\cup Z)$$

$$(tk-1<\lfloor tk\rfloor\leq tk)$$

$$\geq\frac{1}{2}\left[\left(\left(1-\frac{2}{k}\right)^{2}e^{2(t-1)}-e^{-2}\right)f(O\setminus Z)+\left(\frac{1}{2}+\left(\frac{1}{2}-t+\frac{1}{k}\right)\left(1-\frac{2}{k}\right)e^{2(t-1)}\right)f(O)\right.$$

$$\left.-\left(\left(\frac{1}{2}+\frac{1-t}{1-\frac{2}{k}}\right)e^{2(t-1)}-\left(\frac{3}{2}-\frac{1}{k}\right)e^{-2}\right)f(O\cup Z)\right]\qquad\text{(nonnegativity;Lemma A.4)}$$

$$\geq\frac{1}{2}\left[\left(\frac{1}{2}+\left(\frac{3}{2}-t-\frac{1}{k}\right)\left(1-\frac{2}{k}\right)e^{2(t-1)}-e^{-2}\right)f(O)-\left(\left(1-\frac{2}{k}\right)^{2}e^{2(t-1)}-e^{-2}\right)f(O\cap Z)\right.$$

$$\left.-\left(\left(\frac{1}{2}+\frac{1-t}{1-\frac{2}{k}}\right)e^{2(t-1)}-\left(\frac{3}{2}-\frac{1}{k}\right)e^{-2}\right)f(O\cup Z)\right],\qquad\text{(submodularity)}$$

where Inequality $(a)$ and $(b)$ follow from Inequality 6, Lemma C.6 and A.3. $\qquad\square$

*Proof of Theorem 2.1 under matroid constraint.* Let $(f,\mathcal{I})$ be an instance of SM, with optimal solution set $O$. If $f(Z)\geq(0.305-\varepsilon)f(O)$ under matroid constraint, the approximation ratio holds immediately. Otherwise, by Corollary C.1, FASTLS returns a set $Z$ which is an $((1+\varepsilon)\alpha,\alpha)$-guidance set, where $\alpha=0.305-\varepsilon$.

By Lemma C.7 and $Z$ is an $((1+\varepsilon)\alpha,\alpha)$-guidance set with $\alpha=0.305-\varepsilon$,

$$\mathbb{E}\left[f(A_k)\right]\geq\frac{1}{2}\left[\frac{1}{2}+\left(\frac{3}{2}-t-\varepsilon\right)(1-2\varepsilon)e^{2(t-1)}-e^{-2}-(0.305-0.695\varepsilon)\left((1-2\varepsilon)^{2}e^{2(t-1)}-e^{-2}\right)\right.$$

$$\left.-(0.305-\varepsilon)\left(\left(\frac{1}{2}+\frac{1-t}{1-2\varepsilon}\right)e^{2(t-1)}-\left(\frac{3}{2}-\varepsilon\right)e^{-2}\right)\right]f(O)\qquad(\forall k\geq\frac{1}{\varepsilon})$$

$$\geq(0.305-\varepsilon)f(O).\qquad(t=0.559)$$

$$\square$$

# D    Analysis of Deterministic Approximation Algorithm

In this section, we present the pseudocode of deterministic algorithm and its analysis. The organization of this section is as follows: firstly, in Appendix D.1, we introduce the deterministic subroutine, GUIDEDIG-S and GUIDEDIG-M, along with their analysis; then, in Appendix D.2, we provide a randomized version of the deterministic algorithm for analytical purposes; finally, in Appendix D.3, we provide the deterministic algorithm and its theoretical guarantee.

## D.1    Deterministic Subroutines - GUIDEDIG-S and GUIDEDIG-M

Inspired by INTERLACEGREEDY algorithm, a subroutine of INTERPOLATEDGREEDY, proposed by Chen and Kuhnle [11], we introduce guided versions of it for both size and matroid constraints.

---

**Algorithm 8:** A guided INTERLACEGREEDY subroutine for matroid constraints.

---

1 **Procedure** GUIDEDIG-M $(f, \mathcal{I}(\mathcal{M}), Z, G, \ell)$**:**
2     **Input:** oracle $f$, matroid constraint $\mathcal{M}$, guidance set $Z$, starting set $G$ , set size $\ell$
3     **Initialize:** $A, A_1, \ldots, A_\ell \leftarrow \emptyset$
4     **for** $i \leftarrow 1$ *to* $k$ **do**
5         $X_i \leftarrow \{x \in \mathcal{U} \setminus (G \cup A \cup Z) : A + x \in \mathcal{I}(\mathcal{M})\}$
6         $j_i^*, a_i^* \leftarrow \arg\max_{j \in [\ell], x \in X} \Delta(x | G \cup A_j)$
7         $A \leftarrow A + a_i^*, A_{j_i^*} \leftarrow A_{j_i^*} + a_i^*$
8     **end**
9     $\sigma \leftarrow$ a bijection from $G$ to $A$ *s.t.* $(G \cup A_j) \setminus \left( \sum_{x \in A_j} \sigma^{-1}(x) \right)$ is a basis
10     **return** $\left\{ (G \cup A_j) \setminus \left( \sum_{x \in A_j} \sigma^{-1}(x) \right) : 1 \leq j \leq \ell \right\}$

---

The algorithm for the size constraint closely resembles INTERLACEGREEDY in Chen and Kuhnle [11]. Hence, we provide the pseudocode (Alg. 9), guarantees, and analysis in Appendix D.1.1. In this section, we focus on presenting GUIDEDIG-M for matroid constraints as Alg. 8. This algorithm addresses the feasibility issue by incorporating INTERLACEGREEDY into matroid constraints. Moreover, while it compromises the approximation ratio over size constraint to some extent, it no longer has the drawback of low success probability, which the size-constrained version has.

**Algorithm overview.** Under size constraints, INTERPOLATEDGREEDY [11] constructs the solution with $\ell$ iterations, where each iteration involves calling the INTERLACEGREEDY subroutine and adding $k/\ell$ elements into the solution. However, this approach is not applicable to matroid constaint due to the feasibility problem. Consequently, we propose GUIDEDIG-M for matroid constraints designed as follows: 1) consider adding a basis ($k$ elements) $A$ to $\ell$ solution sets, where each addition dominates the gain of a distinct element in $O$; 2) by exchange property, establish a bijection between the basis $A$ and the starting set $G$; 3) delete elements in each solution set that are mapped to by the basis $A$. This procedure avoids the extensive guessing of GUIDEDIG-S for size constraints and reduces the number of potential solutions from $\ell(\ell + 1)$ to $\ell$. We provide the theoretical guarantees and the detailed analysis below.

**Lemma D.1.** Let $O \in \mathcal{I}(\mathcal{M})$, and suppose GUIDEDIG-M(Alg. 8) is called with $(f, \mathcal{M}, Z, G, \ell)$, where $Z \cap G = \emptyset$. Then GUIDEDIG-M outputs $\ell$ candidate sets with $\mathcal{O}(\ell k n)$ queries. Moreover, a randomly selected set $G'$ from the output satisfies that:

1) $\mathbb{E}[f(G')] \geq \left( 1 - \frac{2}{\ell} \right) f(G) + \frac{1}{\ell + 1} \left( 1 - \frac{1}{\ell} \right) f((O \setminus Z) \cup G);$

2) $\mathbb{E}[f(O \cup G')] \geq \left( 1 - \frac{2}{\ell} \right) f(O \cup G) + \frac{1}{\ell} \left( f(O) + f(O \cup (Z \cap G)) - f(O \cup Z) \right);$

3) $\mathbb{E}[f((O \setminus Z) \cup G')] \geq \left( 1 - \frac{2}{\ell} \right) f((O \setminus Z) \cup G) + \frac{1}{\ell} \left( f(O \setminus Z) + f((O \setminus Z) \cup (Z \cap G)) - f(O \cup Z) \right).$

*Proof.* $A = \{a_1^*, a_2^*, \ldots, a_k^*\}$ be the sequence with the order of elements being added. Since $A$ and $O \setminus Z$ are basis of $\mathcal{M}$ (by adding dummy elements into $O \setminus Z$), we can order $O \setminus Z = \{o_1, o_2, \ldots, o_k\}$ *s.t.* for any $1 \leq i \leq k$, $\{a_1^*, \ldots, a_{i-1}^*, o_i\}$ is an independent set. Thus, $o_i \in X_i$. Let $A_j^{(i)}$ be $A_j$ after $i$-th iteration. Therefore, for any $1 \leq j \leq \ell$, by submodularity,

$$\Delta\left( o_i | G \cup A_j \right) \leq \Delta\left( o_i | G \cup A_j^{(i-1)} \right) \leq \Delta\left( a_i^* | G \cup A_{j_i^*}^{(i-1)} \right)$$

$$\Rightarrow f\left( (O \setminus Z) \cup G \cup A_j \right) - f\left( G \cup A_j \right) \leq \sum_{i=1}^{k} \Delta(o_i | G \cup A_j) \leq \sum_{i=1}^{k} \Delta\left( a_i^* | G \cup A_{j_i^*}^{(i-1)} \right) = \sum_{l=1}^{\ell} \Delta(A_l | G)$$

By summing up the above inequality with $1 \leq j \leq \ell$,

$$(\ell + 1) \sum_{j=1}^{\ell} \Delta(A_j | G) \geq \sum_{j=1}^{\ell} f\left( (O \setminus Z) \cup G \cup A_j \right) - \ell f(G)$$

$$\geq (\ell - 1)f\left((O \setminus Z) \cup G\right) + f\left((O \setminus Z) \cup G \cup A\right) - \ell f(G)$$
$$\geq (\ell - 1)f\left((O \setminus Z) \cup G\right) - \ell f(G) \qquad \text{(nonnegativity)}$$

Then, we can prove the first inequality as follows,

$$\mathbb{E}\left[f(G') - f(G)\right] = \frac{1}{\ell} \sum_{j=1}^{\ell} \left(f\left(G \setminus \sigma^{-1}(A_j) \cup A_j\right) - f(G)\right)$$

$$\geq \frac{1}{\ell} \sum_{j=1}^{\ell} \left(\Delta(A_j|G) + f\left(G \setminus \sigma^{-1}(A_j)\right) - f(G)\right)$$

$$\geq \frac{1}{\ell+1}\left(1 - \frac{1}{\ell}\right)f\left((O \setminus Z) \cup G\right) - \frac{1}{\ell+1}f(G) - \frac{1}{\ell}f(G)$$

$$\Rightarrow \qquad \mathbb{E}\left[f(G')\right] \geq \left(1 - \frac{2}{\ell}\right)f(G) + \frac{1}{\ell+1}\left(1 - \frac{1}{\ell}\right)f\left((O \setminus Z) \cup G\right)$$

By submodularity, nonnegativity, and $Z \cap A = \emptyset$, the last two inequalities can be proved as follows,

$$\mathbb{E}\left[f\left(O \cup G'\right)\right] = \frac{1}{\ell} \sum_{j-1}^{\ell} f\left(O \cup \left(G \setminus \sigma^{-1}(A_j) \cup A_j\right)\right)$$

$$\geq \frac{1}{\ell} \sum_{j-1}^{\ell} \left[\Delta(A_j|O \cup G) + f\left(O \cup \left(G \setminus \sigma^{-1}(A_j)\right)\right)\right]$$

$$\geq \frac{1}{\ell}\left(f(O \cup G \cup A) - f(O \cup G) + (\ell - 1)f(O \cup G) + f(O)\right)$$

$$\geq \left(1 - \frac{2}{\ell}\right)f(O \cup G) + \frac{1}{\ell}\left(f(O) + f\left(O \cup (Z \cap G)\right) - f(O \cup Z)\right)$$

$$\mathbb{E}\left[f\left((O \setminus Z) \cup G'\right)\right] = \frac{1}{\ell} \sum_{j-1}^{\ell} f\left((O \setminus Z) \cup \left(G \setminus \sigma^{-1}(A_j) \cup A_j\right)\right)$$

$$\geq \frac{1}{\ell} \sum_{j-1}^{\ell} \left[\Delta(A_j|(O \setminus Z) \cup G) + f\left((O \setminus Z) \cup \left(G \setminus \sigma^{-1}(A_j)\right)\right)\right]$$

$$\geq \frac{1}{\ell}\left(f((O \setminus Z) \cup G \cup A) - f((O \setminus Z) \cup G) + (\ell - 1)f((O \setminus Z) \cup G) + f((O \setminus Z))\right)$$

$$\geq \left(1 - \frac{2}{\ell}\right)f((O \setminus Z) \cup G) + \frac{1}{\ell}\left(f(O \setminus Z) + f\left((O \setminus Z) \cup (Z \cap G)\right) - f(O \cup Z)\right)$$

$\square$

### D.1.1 GUIDEDIG-S and its Analysis

In this section, we provide the pseudocode, guarantees and analysis of GUIDEDIG-S, which highly resembles INTERLACEGREEDY in Chen and Kuhnle [11].

**Lemma D.2.** Let $O \subseteq \mathcal{U}$ be any set of size at most $k$, and suppose GUIDEDIG-S(Alg. 9) is called with $(f, k, Z, G, \ell)$. Then GUIDEDIG-S outputs $\ell(\ell+1)$ candidate sets with $\mathcal{O}(\ell k n)$ queries. Moreover, with a probability of $(\ell+1)^{-1}$, a randomly selected set $A$ from the output satisfies that:

1) $\mathbb{E}\left[f(A)\right] \geq \dfrac{1}{\ell+1}\mathbb{E}\left[f\left((O \setminus Z) \cup A\right)\right] + \dfrac{\ell}{\ell+1}f(G)$;

2) $\mathbb{E}\left[f(O \cup A)\right] \geq \left(1 - \dfrac{1}{\ell}\right)f(O \cup G) + \dfrac{1}{\ell}\left(f\left(O \cup (Z \cap G)\right) - f(O \cup Z)\right)$

3) $\mathbb{E}\left[f\left((O \setminus Z) \cup A\right)\right] \geq \left(1 - \dfrac{1}{\ell}\right)f\left((O \setminus Z) \cup G\right) + \dfrac{1}{\ell}\left(f\left((O \setminus Z) \cup (Z \cap G)\right) - f(O \cup Z)\right)$.

---

**Algorithm 9:** A $(\approx\ell)$-approximation that interlaces $\ell$ greedy procedures together and uses only $1/\ell$ fraction of the budget.

---

1 **Procedure** GUIDEDIG-S $(f, k, Z, G, \ell)$:
2      **Input:** oracle $f$, constraint $k$, guidance set $Z$, starting set $G$, set size $\ell$
3      $\{a_1, \ldots, a_\ell\} \leftarrow$ top $\ell$ elements in $\mathcal{U} \setminus (G \cup Z)$ with respect to marginal gains on $G$
4      **for** $u \leftarrow 0$ *to* $\ell$ *in parallel* **do**
5          **if** $u = 0$ **then**
6              $A_{u,l} \leftarrow G \cup \{a_l\}$, for all $1 \le l \le \ell$
7          **else**
8              $A_{u,l} \leftarrow G \cup \{a_u\}$, for any $1 \le l \le \ell$
9          **end**
10          **for** $j \leftarrow 1$ *to* $k/\ell - 1$ **do**
11              **for** $i \leftarrow 1$ *to* $\ell$ **do**
12                  $x_{j,i} \leftarrow \arg\max_{x \in \mathcal{U} \setminus Z \setminus \left(\cup_{l=1}^\ell A_{u,l}^j\right)} \Delta(x | A_{u,i})$
13                  $A_{u,i} \leftarrow A_{u,i} \cup \{x_{j,i}\}$
14              **end**
15          **end**
16      **end**
17      **return** $\{A_{u,i} : 1 \le i \le \ell, 0 \le u \le \ell\}$

---

*Proof.* Let $o_{\max} = \arg\max_{o \in O \setminus (G \cup Z)} \Delta(o | G)$, and let $\{a_1, \ldots, a_\ell\}$ be the largest $\ell$ elements of $\{\Delta(x | G) : x \in \mathcal{U} \setminus (G \cup Z)\}$, as chosen on Line 3. We consider the following two cases.

**Case** $(O \setminus (G \cup Z)) \cap \{a_1, \ldots, a_\ell\} = \emptyset$**.** Then, $o_{\max} \notin \{a_1, \ldots, a_\ell\}$ which implies that $\Delta(a_u | G) \ge \Delta(o | G)$, for every $1 \le u \le \ell$ and $o \in O \setminus (G \cup Z)$; and, after the first iteration of the **for** loop on Line 10 of Alg. 9, none of the elements in $O \setminus (G \cup Z)$ is added into any of $\{A_{0,i}\}_{i=1}^\ell$. We will analyze the iteration of the **for** loop on Line 4 with $u = 0$.

Since none of the elements in $O \setminus (G \cup Z)$ is added into the collection when $j = 0$, we can order $O \setminus (G \cup Z) = \{o_1, o_2, \ldots\}$ such that the first $\ell$ elements are not selected in any set before we get to $j = 1$, the next $\ell$ elements are not selected in any set before we get to $j = 2$, and so on. Let $i \in \{1, \ldots, \ell\}$. Let $A_{0,i}^j$ be the value of $A_{0,i}$ after $j$ elements are added into it, and define $A_{0,i} = A_{0,i}^{k/\ell}$, the final value. Finally, denote by $\delta_j$ the value $\Delta\left(x_{j,i} | A_{0,i}^j\right)$. Then,

$$f\left((O \setminus Z) \cup A_{0,i}\right) - f\left(A_{0,i}\right) \le \sum_{o \in O \setminus (A_{0,i} \cup Z)} \Delta(o | A_{0,i}) \qquad \text{(submodularity)}$$

$$\le \sum_{o \in O \setminus (G \cup Z)} \Delta(o | A_{0,i}) \qquad (G \subseteq A_{0,i})$$

$$\le \sum_{l=1}^{\ell} \Delta\left(o_l | A_{0,i}^0\right) + \sum_{l=\ell+1}^{2\ell} \Delta\left(o_l | A_{0,i}^1\right) + \ldots \qquad \text{(submodularity)}$$

$$\le \ell \sum_{j=1}^{k/\ell} \delta_j = \ell(f\left(A_{0,i}\right) - f\left(G\right)),$$

where the last inequality follows from the ordering of $O$ and the selection of elements into the sets. By summing up the above inequality with all $1 \le i \le \ell$, it holds that,

$$\mathbb{E}\left[f(A)\right] = \frac{1}{\ell} \sum_{i=1}^{\ell} f\left(A_{0,i}\right) \ge \frac{1}{\ell(\ell+1)} \sum_{i=1}^{\ell} f((O \setminus Z) \cup A_{0,i}) + \frac{\ell}{\ell+1} f(G)$$

$$= \frac{1}{\ell+1} \mathbb{E}\left[f\left((O \setminus Z) \cup A\right)\right] + \frac{\ell}{\ell+1} f\left(G\right),$$

Since $A_{0,i_1} \cap A_{0,i_2} = G$ for any $1 \le i_1 \ne i_2 \le \ell$, and each $x_{j,i}$ is selected outside of $Z$, by repeated application of submodularity, it can be shown that

$$\mathbb{E}\left[f\left((O \setminus Z) \cup A\right)\right] = \frac{1}{\ell}\sum_{i=1}^{\ell} f((O \setminus Z) \cup A_{0,i})$$

$$\ge \left(1 - \frac{1}{\ell}\right) f\left((O \setminus Z) \cup G\right) + \frac{1}{\ell} f\left((O \setminus Z) \cup \left(\cup_{i=1}^{\ell} A_{0,i}\right)\right)$$

$$\ge \left(1 - \frac{1}{\ell}\right) f\left((O \setminus Z) \cup G\right) + \frac{1}{\ell}(f\left((O \setminus Z) \cup (Z \cap G)\right) - f(O \cup Z))$$

$$\mathbb{E}\left[f\left(O \cup A\right)\right] = \frac{1}{\ell}\sum_{i=1}^{\ell} f(O \cup A_{0,i}) \ge \left(1 - \frac{1}{\ell}\right) f\left(O \cup G\right) + \frac{1}{\ell} f\left(O \cup \left(\cup_{i=1}^{\ell} A_{0,i}\right)\right)$$

$$\ge \left(1 - \frac{1}{\ell}\right) f\left(O \cup G\right) + \frac{1}{\ell}\left(f\left(O \cup (Z \cap G)\right) - f\left(O \cup Z\right)\right).$$

Therefore, if we select a random set from $\{A_{0,i} : 1 \le i \le \ell\}$, the three inequalities in the Lemma hold and we have probability $1/(\ell + 1)$ of this happening.

**Case** $(O \setminus (G \cup Z)) \cap \{a_1, \ldots, a_\ell\} \ne \emptyset$. Then $o_{\max} \in \{a_1, \ldots, a_\ell\}$, so $a_u = o_{\max}$, for some $u \in 1, \ldots, \ell$. We analyze the iteration $u$ of the **for** loop on Line 4. Similar to the previous case, let $i \in \{1, \ldots, \ell\}$, define $A_{u,i}^j$ be the value of $A_{u,i}$ after we add $j$ elements into it, and we will use $A_{u,i}$ for $A_{u,i}^{k/\ell}$, Also, let $\delta_j = \Delta\left(x_{j,i} | A_{u,i}^{j-1}\right)$. Finally, let $x_{1,i} = a_u$ and observe $A_{u,i}^1 = G \cup \{a_u\}$, for any $i \in \{1, \ldots, \ell\}$.

Then, we can order $O \setminus G = \{o_1, o_2, \ldots\}$ such that: 1) for the first $\ell$ elements $\{o_l\}_{l=1}^{\ell}$, $\Delta(o_l | G) \le \Delta(o_{\max} | G) = \delta_1$; 2) the next $\ell$ elements $\{o_l\}_{l=\ell+1}^{2\ell}$ are not selected by any set before we get to $j = 2$, which implies that $\Delta\left(o_l | A_{u,i}^1\right) \le \delta_2$, and so on. Therefore, analogous to the the previous case, we have that

$$\mathbb{E}\left[f\left(A\right)\right] \ge \frac{1}{\ell+1}\mathbb{E}\left[f\left((O \setminus Z) \cup A\right)\right] + \frac{\ell}{\ell+1}f\left(G\right) \tag{7}$$

Since, $A_{u,i_1} \cap A_{u,i_2} = G \cup \{a_u\}$ for any $1 \le i_1 \ne i_2 \le \ell$, $a_u \in O \setminus Z$, and each $x_{j,i}$ is selected outside of $Z$, by submodularity and nonnegativity of $f$, it holds that

$$\mathbb{E}\left[f\left((O \setminus Z) \cup A\right)\right] = \frac{1}{\ell}\sum_{i=1}^{\ell} f((O \setminus Z) \cup A_{u,i})$$

$$\ge \left(1 - \frac{1}{\ell}\right) f\left((O \setminus Z) \cup G\right) + \frac{1}{\ell} f\left((O \setminus Z) \cup \left(\cup_{i=1}^{\ell} A_{u,i}\right)\right)$$

$$\ge \left(1 - \frac{1}{\ell}\right) f\left((O \setminus Z) \cup G\right) + \frac{1}{\ell}(f\left((O \setminus Z) \cup (Z \cap G)\right) - f(O \cup Z))$$

$$\mathbb{E}\left[f\left(O \cup A\right)\right] = \frac{1}{\ell}\sum_{i=1}^{\ell} f(O \cup A_{u,i}) \ge \left(1 - \frac{1}{\ell}\right) f\left(O \cup G\right) + \frac{1}{\ell} f\left(O \cup \left(\cup_{i=1}^{\ell} A_{u,i}\right)\right)$$

$$\ge \left(1 - \frac{1}{\ell}\right) f\left(O \cup G\right) + \frac{1}{\ell}\left(f\left(O \cup (Z \cap G)\right) - f\left(O \cup Z\right)\right).$$

Therefore, if we select a random set from $\{A_{u,i} : 1 \le i \le \ell\}$, the three inequalities in the lemma holds, and this happens with probability $(\ell + 1)^{-1}$. $\qquad\square$

### D.2 Randomized Version of our Deterministic Algorithm

In this section, we provide the randomized version (Alg 10) of our deterministic algorithm (Alg. 11, provided in Appendix D). The deterministic version simply evaluates all possible paths and returns the best solution. In the following, we provide the theoretical guarantee and its analysis under different constraints.

---

**Algorithm 10:** The randomized algorithm suitable for derandomization.

---

1  **Input:** oracle $f$, constraint $\mathcal{I}$, an approximation result $Z_0$, switch point $t$, error rate $\varepsilon$

2  $Z \leftarrow \textsc{FastLS}(f, \mathcal{I}, Z_0, \varepsilon)$

3  **Initialize** $\ell \leftarrow \frac{10}{9\varepsilon}, A_0 \leftarrow \emptyset$

4  **if** $\mathcal{I}$ *is a size constraint* **then**

5     **for** $i \leftarrow 1$ *to* $\ell$ **do**

6         **if** $i \leq t\ell$ **then** $A_i \leftarrow$ a random set in $\textsc{GuidedIG-S}(f, \mathcal{I}, Z, A_{i-1}, \ell)$

7         **else** $A_i \leftarrow$ a random set in $\textsc{GuidedIG-S}(f, \mathcal{I}, \emptyset, A_{i-1}, \ell)$

8     **end**

9  **else**

10    **for** $i \leftarrow 1$ *to* $\ell$ **do**

11       **if** $i \leq t\ell$ **then** $A_i \leftarrow$ a random set in $\textsc{GuidedIG-M}(f, \mathcal{I}, Z, A_{i-1}, \ell)$

12       **else** $A_i \leftarrow$ a random set in $\textsc{GuidedIG-M}(f, \mathcal{I}, \emptyset, A_{i-1}, \ell)$

13    **end**

14  **end**

15  **return** $A^* \leftarrow \arg\max\{f(Z), f(A_\ell)\}$

---

**Theorem D.3.** Let $(f, \mathcal{I})$ be an instance of SM, with the optimal solution set $O$. Algorithm 10 achieves an expected $(0.385 - \varepsilon)$ approximation ratio with $(\ell + 1)^{-\ell}$ success probability and input $t = 0.372$ under size constraint, where $\ell = \frac{10}{9\varepsilon}$. Moreover, it achieves an expected $(0.305 - \varepsilon)$ approximation ratio with $t = 0.559$ under matroid constraint. The query complexity of the algorithm is $\mathcal{O}(kn/\varepsilon)$.

### D.2.1   Size constraints

By Lemma D.2 in Appendix D.1.1 and the closed form for a recurrence provided in Lemma A.3, the following corollary holds,

**Corollary D.1.** After iteration $i$ of the for loop in Alg. 10, the following inequalities hold with a probability of $(\ell + 1)^{-i}$

$$\mathbb{E}[f(A_i)] \geq \frac{\ell}{\ell+1}\mathbb{E}[f(A_{i-1})] + \frac{1}{\ell+1}\left(f(O \setminus Z) - \left(1 - \left(1 - \frac{1}{\ell}\right)^i\right)f(O \cup Z)\right), \qquad 1 \leq i \leq t\ell$$

$$\mathbb{E}[f(A_i)] \geq \frac{\ell}{\ell+1}\mathbb{E}[f(A_{i-1})] + \frac{1}{\ell+1}\left(1 - \frac{1}{\ell}\right)^{i - \lfloor t\ell \rfloor}\left(f(O) - \left(1 - \left(1 - \frac{1}{\ell}\right)^{\lfloor t\ell \rfloor}\right)f(O \cup Z)\right), \quad t\ell < i \leq \ell$$

*Proof of approximation ratio.* If $f(Z) \geq (0.385 - \varepsilon)f(O)$, the approximation ratio holds immediately. So, we analyze the case $f(Z) < (0.385 - \varepsilon)f(O)$ in the following.

Recall in Corollary C.1 that $Z$ is a $(1 + \varepsilon)\alpha, \alpha)$-guidance set, it holds that $f(O \cup Z) + f(O \cap Z) < (0.77 - 1.615\varepsilon)f(O)$ and $f(O \cap Z) < (0.385 - 0.615\varepsilon)f(O)$.

By repeatedly implementing Lemma A.3 with the recursion in Corollary D.1, it holds that

$$\mathbb{E}\left[f\left(A_{\lfloor t\ell \rfloor}\right)\right] \geq \left(1 - \left(1 - \frac{1}{\ell+1}\right)^{\lfloor t\ell \rfloor}\right)(f(O \setminus Z) - f(O \cup Z)) + \frac{\lfloor t\ell \rfloor}{\ell+1}\left(1 - \frac{1}{\ell}\right)^{\lfloor t\ell \rfloor}f(O \cup Z)$$

$$\geq \left(1 - \left(1 - \frac{1}{\ell+1}\right)^{\lfloor t\ell \rfloor}\right)(f(O) - f(O \cap Z) - f(O \cup Z)) + \frac{\lfloor t\ell \rfloor}{\ell+1}\left(1 - \frac{1}{\ell}\right)^{\lfloor t\ell \rfloor}f(O \cup Z)$$

$$\text{(submodularity)}$$

$$\mathbb{E}[f(A_\ell)] \geq \left(1 - \frac{1}{\ell+1}\right)^{\ell - \lfloor t\ell \rfloor}\mathbb{E}\left[f\left(A_{\lfloor t\ell \rfloor}\right)\right] + \frac{\ell - \lfloor t\ell \rfloor}{\ell+1}\left(1 - \frac{1}{\ell}\right)^{\ell - \lfloor t\ell \rfloor}\left(f(O) - \left(1 - \left(1 - \frac{1}{\ell}\right)^{\lfloor t\ell \rfloor}\right)f(O \cup Z)\right)$$

$$\geq \left(\left(1 - \frac{1}{\ell+1}\right)^{(1-t)\ell+1} - \left(1 - \frac{1}{\ell+1}\right)^{\ell}\right)(f(O) - f(O \cap Z) - f(O \cup Z))$$

$$+ \frac{t\ell}{\ell+1} \left(1 - \frac{1}{\ell+1}\right)^{(1-t)\ell+1} \left(1 - \frac{1}{\ell}\right)^{t\ell} f(O \cup Z)$$

$$+ (1-t)\left(1 - \frac{1}{\ell+1}\right)\left(1 - \frac{1}{\ell}\right)^{(1-t)\ell+1}\left(f(O) - \left(1 - \left(1 - \frac{1}{\ell}\right)^{t\ell-1}\right) f(O \cup Z)\right)$$
$$(t\ell - 1 < \lfloor t\ell \rfloor \leq t\ell)$$

$$= \left(\left(1 - \frac{1}{\ell+1}\right)\left(1 - \frac{1}{\ell+1}\right)^{(1-t)\ell} - \left(1 + \frac{1}{\ell}\right)\left(1 - \frac{1}{\ell+1}\right)^{\ell+1}\right)(f(O) - f(O \cap Z) - f(O \cup Z))$$

$$+ t \cdot \frac{\ell-1}{\ell+1}\left(1 - \frac{1}{\ell+1}\right)\left(1 - \frac{1}{\ell+1}\right)^{(1-t)\ell}\left(1 - \frac{1}{\ell}\right)^{t\ell-1} f(O \cup Z)$$

$$+ (1-t)\left(1 - \frac{1}{\ell+1}\right)^2 \left[\left(1 - \frac{1}{\ell}\right)\left(1 - \frac{1}{\ell}\right)^{(1-t)\ell-1} f(O)\right.$$
$$\left. - \left(1 - \frac{1}{\ell}\right)^{(1-t)\ell} f(O \cup Z) + \left(1 - \frac{1}{\ell}\right)\left(1 - \frac{1}{\ell}\right)^{\ell-1} f(O \cup Z)\right]$$

$$\geq \left(\left(1 - \frac{1}{\ell}\right)e^{t-1} - e^{-1}\right)(f(O) - f(O \cap Z) - f(O \cup Z)) + t \cdot \left(1 - \frac{1}{\ell}\right)^3 e^{-1} f(O \cup Z)$$

$$+ (1-t)\left(1 - \frac{1}{\ell}\right)^2 \left(\left(1 - \frac{1}{\ell}\right)e^{t-1}f(O) - e^{t-1}f(O \cup Z) + \left(1 - \frac{1}{\ell}\right)e^{-1}f(O \cup Z)\right)$$
$$(f(O) - f(O \cap Z) - f(O \cup Z) > 0; \text{ nonnegativity; Lemma A.4})$$

$$\geq \left[c(c^2(1-t)+1)e^{t-1} - e^{-1}\right]f(O) - \left[c(c(1-t)+1)e^{t-1} - (c^3+1)e^{-1}\right](f(O \cup Z) + f(O \cap Z))$$
$$- \left[c^3 e^{-1} - c^2(1-t)e^{t-1}\right]f(O \cap Z) \qquad \left(\text{Let } c = 1 - \frac{9\varepsilon}{10} = 1 - \frac{1}{\ell}\right)$$

$$\geq \left[c(c^2(1-t)+1)e^{t-1} - e^{-1}\right]f(O) - \left[c(c(1-t)+1)e^{t-1} - (c^3+1)e^{-1}\right](0.77 - 1.615\varepsilon)f(O)$$
$$- \left[c^3 e^{-1} - c^2(1-t)e^{t-1}\right](0.385 - 0.615\varepsilon)f(O)$$
$$(f(O \cup Z) + f(O \cap Z) < (0.77 - 1.615\varepsilon)f(O); f(O \cap Z) < (0.385 - 0.615\varepsilon)f(O))$$

$$\geq (0.385 - \varepsilon)f(O) \qquad (0 < \varepsilon < 0.385; t = 0.372)$$

$$\square$$

### D.2.2 Matroid Constraints

By Lemma D.1 in Appendix D.1 and the closed form for a recurrence provided in Lemma A.3, the following corollary holds,

**Corollary D.2.** After iteration $i$ of the for loop in Alg. 10, the following inequalities hold,

$$\mathbb{E}[f(A_i)] \geq \left(1 - \frac{2}{\ell}\right)\mathbb{E}[f(A_{i-1})]$$
$$+ \frac{1}{\ell+1}\left(1 - \frac{1}{\ell}\right)\left(f(O \setminus Z) - \frac{1}{2}\left(1 - \left(1 - \frac{2}{\ell}\right)^{i-1}\right)f(O \cup Z)\right), 1 \leq i \leq t\ell$$

$$\mathbb{E}[f(A_i)] \geq \left(1 - \frac{2}{\ell}\right)\mathbb{E}[f(A_{i-1})]$$
$$+ \frac{1}{\ell+1}\left(1 - \frac{1}{\ell}\right)\left(\frac{1}{2}\left(1 + \left(1 - \frac{2}{\ell}\right)^{i-t\ell}\right)f(O) - \frac{1}{2}\left(\left(1 - \frac{2}{\ell}\right)^{i-t\ell} - \left(1 - \frac{2}{\ell}\right)^i\right)f(O \cup Z)\right), t\ell < i \leq \ell$$

*Proof of approximation ratio.* If $f(Z) \geq (0.305 - \varepsilon)f(O)$, the approximation ratio holds immediately. So, we analyze the case $f(Z) < (0.305 - \varepsilon)f(O)$ in the following.

Recall that $Z$ is a $(1+\varepsilon)\alpha, \alpha)$-guidance set in Corollary C.1, it holds that $f(O \cup Z) + f(O \cap Z) < (0.61 - 1.695\varepsilon)f(O)$ and $f(O \cap Z) < (0.305 - 0.695\varepsilon)f(O)$.

By repeatedly implementing Lemma A.3 with the recursion in Corollary D.2, it holds that

$$\mathbb{E}\left[f\left(A_{\lfloor t\ell\rfloor}\right)\right] \geq \frac{\ell-1}{2(\ell+1)}\left[\left(1-\left(1-\frac{2}{\ell}\right)^{\lfloor t\ell\rfloor}\right)\left(f\left(O\setminus Z\right)-\frac{1}{2}f\left(O\cup Z\right)\right)+t\left(1-\frac{2}{\ell}\right)^{\lfloor t\ell\rfloor-1}f\left(O\cup Z\right)\right]$$

$$\geq \frac{\ell-1}{2(\ell+1)}\left[\left(1-\left(1-\frac{2}{\ell}\right)^{\lfloor t\ell\rfloor}\right)\left(f\left(O\right)-f\left(O\cap Z\right)-\frac{1}{2}f\left(O\cup Z\right)\right)+t\left(1-\frac{2}{\ell}\right)^{\lfloor t\ell\rfloor-1}f\left(O\cup Z\right)\right]$$

$$\text{(submodularity)}$$

$$\mathbb{E}\left[f\left(A_\ell\right)\right] \geq \left(1-\frac{2}{\ell}\right)^{\ell-\lfloor t\ell\rfloor}\mathbb{E}\left[f\left(A_{\lfloor t\ell\rfloor}\right)\right]+\frac{\ell-1}{2(\ell+1)}\left[\left(\frac{1}{2}+\left(\frac{1}{2}-t+\frac{1}{\ell}\right)\left(1-\frac{2}{\ell}\right)^{\ell-\lfloor t\ell\rfloor-1}\right)f(O)\right.$$

$$\left.-(1-t)\left(\left(1-\frac{2}{\ell}\right)^{\ell-\lfloor t\ell\rfloor-1}-\left(1-\frac{2}{\ell}\right)^{\ell-1}\right)f(O\cup Z)\right]$$

$$\geq \frac{\ell-1}{2(\ell+1)}\left[\left(\frac{1}{2}+\left(\frac{3}{2}-t-\frac{1}{\ell}\right)\left(1-\frac{2}{\ell}\right)^{\ell-t\ell}-\left(1-\frac{2}{\ell}\right)^{\ell}\right)f\left(O\right)\right.$$

$$-\left(\left(1-\frac{2}{\ell}\right)^{(1-t)\ell}-\left(1-\frac{2}{\ell}\right)^{\ell}\right)f\left(O\cap Z\right)$$

$$\left.-\left(\left(\frac{1}{2}+\frac{1-t}{1-\frac{2}{\ell}}\right)\left(1-\frac{2}{\ell}\right)^{\ell-t\ell}-\left(\frac{3}{2}-\frac{1}{\ell}\right)\left(1-\frac{2}{\ell}\right)^{\ell-1}\right)f\left(O\cup Z\right)\right]$$

$$(t\ell-1<\lfloor t\ell\rfloor\leq t\ell)$$

$$\geq \frac{\ell-1}{2(\ell+1)}\left[\left(\frac{1}{2}+\left(\frac{3}{2}-t-\frac{1}{\ell}\right)\left(1-\frac{2}{\ell}\right)e^{2(t-1)}-e^{-2}\right)f\left(O\right)-\left(e^{2(t-1)}-\left(1-\frac{2}{\ell}\right)e^{-2}\right)f\left(O\cap Z\right)\right.$$

$$\left.-\left(\left(\frac{1}{2}+\frac{1-t}{1-\frac{2}{\ell}}\right)e^{2(t-1)}-\left(\frac{3}{2}-\frac{1}{\ell}\right)e^{-2}\right)f\left(O\cup Z\right)\right]\qquad\text{(Lemma A.4; nonnegativity)}$$

$$\geq \frac{\ell-1}{2(\ell+1)}\left[\left(\frac{1}{2}+\left(\frac{3}{2}-t-\frac{1}{\ell}\right)\left(1-\frac{2}{\ell}\right)e^{2(t-1)}-e^{-2}\right)f\left(O\right)-\left(\left(\frac{1}{2}+\frac{1}{\ell}\right)e^{-2}-\left(\frac{1}{2}-t\right)e^{2(t-1)}\right)f\left(O\cap Z\right)\right.$$

$$\left.-\left(\left(\frac{1}{2}+\frac{1-t}{1-\frac{2}{\ell}}\right)e^{2(t-1)}-\left(\frac{3}{2}-\frac{1}{\ell}\right)e^{-2}\right)(f\left(O\cup Z\right)+f\left(O\cap Z\right))\right]$$

$$\geq \frac{\ell-1}{2(\ell+1)}\left[\frac{1}{2}+\left(\frac{3}{2}-t-\frac{1}{\ell}\right)\left(1-\frac{2}{\ell}\right)e^{2(t-1)}-e^{-2}-(0.305-0.695\varepsilon)\left(\left(\frac{1}{2}+\frac{1}{\ell}\right)e^{-2}-\left(\frac{1}{2}-t\right)e^{2(t-1)}\right)\right.$$

$$\left.-(0.61-1.695\varepsilon)\left(\left(\frac{1}{2}+\frac{1-t}{1-\frac{2}{\ell}}\right)e^{2(t-1)}-\left(\frac{3}{2}-\frac{1}{\ell}\right)e^{-2}\right)\right]f\left(O\right)$$

$$\geq (0.305-\varepsilon)f\left(O\right)\qquad\qquad\left(\ell=\frac{10}{9\varepsilon};0<\varepsilon<0.305;t=0.559\right)$$

$$\square$$

### D.3 Derandomize Alg. 10 in Section 2.3

In this section, we present the deranmized version of Alg. 10, which simply evaluates all possible paths and returns the best solution. We reiterate the guarantee as follows.

**Theorem 2.4.** Let $(f,k)$ be an instance of SM, with the optimal solution set $O$. Alg. 11 achieves a deterministic $(0.385-\varepsilon)$ approximation ratio with $t=0.372$, and a deterministic $(0.305-\varepsilon)$ approximation ratio with $t=0.559$. The query complexity of the algorithm is $\mathcal{O}\left(kn\ell^{2\ell-1}\right)$ where $\ell=\frac{10}{9\varepsilon}$.

## E  Proofs for Nearly Linear Time Deterministic Algorithm in Section 3

In this section, we provide the pseudocode and additional analysis of our nearly linear time deterministic algorithm introduced in Section 3. We organize this section as follows: in Appendix E.1, we

**Algorithm 11:** Deterministic combinatorial approximation algorithm with the same ratio as Alg. 2

---

**1 Input:** oracle $f$, constraint $\mathcal{I}$, an approximation result $Z_0$, switch point $t$, error rate $\varepsilon$

**2** $Z \leftarrow \text{FastLS}(f, \mathcal{I}, Z_0, \varepsilon)$

**3 Initialize** $\ell \leftarrow \frac{10}{9\varepsilon}, G_0 \leftarrow \{\emptyset\}$

**4 if** $\mathcal{I}$ *is a size constraint* **then**

**5** $\quad$ **for** $i \leftarrow 1$ *to* $\ell$ **do**

**6** $\quad\quad$ $G_i \leftarrow \emptyset$

**7** $\quad\quad$ **for** $A_{i-1} \in G_{i-1}$ **do**

**8** $\quad\quad\quad$ **if** $i \leq t\ell$ **then**

**9** $\quad\quad\quad\quad$ $G_i \leftarrow G_i \cup \text{GuidedIG-S}(f, k, Z, A_{i-1}, \ell)$

**10** $\quad\quad\quad$ **else**

**11** $\quad\quad\quad\quad$ $G_i \leftarrow G_i \cup \text{GuidedIG-S}(f, k, \emptyset, A_{i-1}, \ell)$

**12** $\quad\quad\quad$ **end**

**13** $\quad\quad$ **end**

**14** $\quad$ **end**

**15 else**

**16** $\quad$ **for** $i \leftarrow 1$ *to* $\ell$ **do**

**17** $\quad\quad$ $G_i \leftarrow \emptyset$

**18** $\quad\quad$ **for** $A_{i-1} \in G_{i-1}$ **do**

**19** $\quad\quad\quad$ **if** $i \leq t\ell$ **then**

**20** $\quad\quad\quad\quad$ $G_i \leftarrow G_i \cup \text{GuidedIG-M}(f, k, Z, A_{i-1}, \ell)$

**21** $\quad\quad\quad$ **else**

**22** $\quad\quad\quad\quad$ $G_i \leftarrow G_i \cup \text{GuidedIG-M}(f, k, \emptyset, A_{i-1}, \ell)$

**23** $\quad\quad\quad$ **end**

**24** $\quad\quad$ **end**

**25** $\quad$ **end**

**26 end**

**27 return** $A^* \leftarrow \arg\max\{f(Z), f(A_\ell) : A_\ell \in G_\ell\}$

---

provide a speedup version of GuidedIG-S which queries $\mathcal{O}(n \log(k))$ times; in Appendix E.2, we analyze the subroutine, Prune; at last, in Appendix E.3, we provide the pseudocode of nearly linear time deterministic algorithm and its analysis.

### E.1 GuidedIG-S Speedup

In this section, we provide the algorithm ThreshGuidedIG, which combines the guiding and descending threshold greedy techniques with InterlaceGreedy [11].

**Lemma E.1.** Let $O \subseteq \mathcal{U}$ be any set of size at most $k$, and suppose ThreshGuidedIG(Alg. 12) is called with $(f, k, \overline{Z}, G, \ell)$. Then ThreshGuidedIG outputs $\ell(\ell + 1)$ candidate sets with $\mathcal{O}(\ell^2 n \log(k)/\varepsilon)$ queries. Moreover, with a probability of $(\ell + 1)^{-1}$, a randomly selected set $A$ from the output satisfies that:

1) $\left( \frac{\ell}{1 - \varepsilon} + 1 \right) \mathbb{E}\left[ f(A) \right] \geq \mathbb{E}\left[ f((O \setminus Z) \cup A) \right] + \frac{\ell}{1 - \varepsilon} f(G) - \varepsilon f(O);$

2) $\mathbb{E}\left[ f(O \cup A) \right] \geq \left( 1 - \frac{1}{\ell} \right) f(O \cup G) + \frac{1}{\ell} \left( f(O \cup (Z \cap G)) - f(O \cup Z) \right)$

3) $\mathbb{E}\left[ f((O \setminus Z) \cup A) \right] \geq \left( 1 - \frac{1}{\ell} \right) f((O \setminus Z) \cup G) + \frac{1}{\ell} (f((O \setminus Z) \cup (Z \cap G)) - f(O \cup Z)).$

*Proof.* Let $o_{\max} = \arg\max_{o \in O \setminus (G \cup Z)} \Delta(o|G)$, and let $\{a_1, \ldots, a_\ell\}$ be the largest $\ell$ elements of $\{\Delta(x|G) : x \in \mathcal{U} \setminus (G \cup Z)\}$, as chosen on Line 13. We consider the following two cases.

**Case** $(O \setminus (G \cup Z)) \cap \{a_1, \ldots, a_\ell\} = \emptyset$. Then, $o_{\max} \notin \{a_1, \ldots, a_\ell\}$ which implies that $\Delta(a_u|G) \geq \Delta(o|G)$, for every $1 \leq u \leq \ell$ and $o \in O \setminus (G \cup Z)$; and, after the first iteration of the **while** loop on

**Algorithm 12:** A guided INTERLACEGREEDY subroutine with descending threshold technique for size constraints.

1 **Procedure** THRESHGUIDEDIG $(f, k, Z, G, \ell, \varepsilon)$:
2      **Input:** oracle $f$, constraint $k$, guidance set $Z$, starting set $G$, set size $\ell$, error $\varepsilon > 0$
3      $\{a_1, \ldots, a_\ell\} \leftarrow$ top $\ell$ elements in $\mathcal{U} \setminus (G \cup Z)$ with respect to marginal gains on $G$
4      **for** $u \leftarrow 0$ *to* $\ell$ *in parallel* **do**
5          **if** $u = 0$ **then**
6              $M \leftarrow \Delta(a_\ell | G)$
7              $A_{u,l} \leftarrow G \cup \{a_l\}$, for any $l \in [\ell]$
8          **else**
9              $M \leftarrow \Delta(a_u | G)$
10              $A_{u,l} \leftarrow G \cup \{a_u\}$, for any $l \in [\ell]$
11          **end**
12          $\tau_l \leftarrow M$, $I_l \leftarrow$ **true**, for any $l \in [\ell]$
13          **while** $\vee_{l=1}^{\ell} I_l$ **do**
14              **for** $i \leftarrow 1$ *to* $\ell$ **do**
15                  **if** $I_i$ **then**
16                      $V \leftarrow \mathcal{U} \setminus \left( \cup_{l=1}^{\ell} A_{u,l} \cup Z \right)$
17                      $A_{u,i}, \tau_i \leftarrow \text{ADD}(f, V, A_{u,i}, \varepsilon, \tau_i, \frac{\varepsilon M}{k})$
18                      **if** $|A_{u,i} \setminus G| = k/\ell \vee \tau_i < \frac{\varepsilon M}{k}$ **then** $I_i \leftarrow$ **false**
19                  **end**
20              **end**
21          **end**
22          $G_m \leftarrow G_m \cup \{A_{u,1}, A_{u,2}, \ldots, A_{u,\ell}\}$
23      **end**
24      **return** $S^* \leftarrow \arg\max\{f(S_\ell) : S_\ell \in G_\ell\}$
25 **Procedure** ADD $(f, V, A, \varepsilon, \tau, \tau_{\min})$:
26      **Input:** oracle $f$, candidate set $V$, solution set $A$, $\varepsilon$, threshold $\tau$, and its lower bound $\tau_{\min}$
27      **while** $\tau \geq \tau_{min}$ **do**
28          **for** $x \in V$ **do**
29              **if** $\Delta(x|A) \geq \tau$ **then return** $A \leftarrow A \cup \{x\}, \tau$
30              $\tau \leftarrow (1 - \varepsilon)\tau$
31          **end**
32      **end**
33      **return** $A, \tau$

Line 13, none of the elements in $O \setminus (G \cup Z)$ is added into any of $\{A_{0,i}\}_{i=1}^{\ell}$. We will analyze the iteration of the **for** loop on Line 4 with $u = 0$.

For any $l \in [\ell]$, let $A_{0,l}^{(j)}$ be $A_{0,l}$ after we add $j$ elements into it, $\tau_l^{(j)}$ be $\tau_l$ when we adopt $j$-th elements into $A_{0,l}$, and $\tau_l^{(1)} = M$. By Line 7, it holds that $A_{0,l}^{(1)} = G \cup \{a_l\}$. Since $(O \setminus (G \cup Z)) \cup \{a_1, \ldots, a_\ell\} = \emptyset$, and we add elements to each set in turn, we can order $O \setminus (G \cup Z) = \{o_1, o_2, \ldots\}$ such that the first $\ell$ elements are not selected by any set before we get $A_{0,l}^{(1)}$, the next $\ell$ elements are not selected in any set before we get $A_{0,l}^{(2)}$, and so on. Therefore, for any $j \leq |A_{0,l} \setminus G|$ and $\ell(j-1) + 1 \leq i \leq \ell j$, $o_i$ are filtered out by $A_{0,l}$ with threshold $\tau_l^{(j)}/(1 - \varepsilon)$, which follows that $\Delta\left(o_i | A_{0,l}^{(j)}\right) < \tau_l^{(j)}/(1-\varepsilon) \leq (f(A_{0,l}^{(j)}) - f(A_{0,l}^{(j-1)}))/(1-\varepsilon)$; for any $\ell|A_{0,l} \setminus G| < i \leq |O \setminus (G \cup Z)|$, $o_i$ are filtered out by $A_{0,l}$ with threshold $\frac{\varepsilon M}{k}$, which follows that $\Delta(o_i | A_{0,l}) < \varepsilon M / k$. Thus,

$$f\left((O \setminus Z) \cup A_{0,l}\right) - f\left(A_{0,l}\right) \leq \sum_{o \in O \setminus A_{0,l}} \Delta(o | A_{0,l}) \qquad \text{(submodularity)}$$

$$\leq \sum_{o \in O \setminus (G \cup Z)} \Delta(o | A_{0,l}) \qquad (G \subseteq A_{0,l})$$

$$\leq \sum_{i=1}^{\ell} \Delta\left(o_i | A_{0,l}^{(1)}\right) + \sum_{i=\ell+1}^{2\ell} \Delta\left(o_i | A_{0,l}^{(2)}\right) + \ldots + \sum_{i > \ell | A_{0,l} \setminus G|} \Delta(o_i | A_{0,l}) \qquad \text{(submodularity)}$$

$$\leq \ell \cdot \frac{f(A_{0,l}) - f(G)}{1 - \varepsilon} + \varepsilon M$$

$$\leq \ell \cdot \frac{f(A_{0,l}) - f(G)}{1 - \varepsilon} + \varepsilon f(O).$$

By summing up the above inequality with all $1 \leq l \leq \ell$, it holds that

$$\left(\frac{\ell}{1 - \varepsilon} + 1\right) \mathbb{E}\left[f(A)\right] \geq \mathbb{E}\left[f((O \setminus Z) \cup A)\right] + \frac{\ell}{1 - \varepsilon} f(G) - \varepsilon f(O). \qquad (8)$$

Since $A_{u,l_1} \cap A_{u,l_2} = G$ for any $1 \leq l_1 \neq l_2 \leq \ell$ it holds that $((O \setminus Z) \cup A_{u,l_1}) \cap ((O \setminus Z) \cup A_{u,l_2}) = (O \setminus Z) \cup G$. By repeated application of submodularity, it holds that

$$\mathbb{E}\left[f((O \setminus Z) \cup A)\right] = \frac{1}{\ell} \sum_{i=1}^{\ell} f((O \setminus Z) \cup A_{0,i})$$

$$\geq \left(1 - \frac{1}{\ell}\right) f((O \setminus Z) \cup G) + \frac{1}{\ell} f\left((O \setminus Z) \cup \left(\cup_{i=1}^{\ell} A_{0,i}\right)\right)$$

$$\geq \left(1 - \frac{1}{\ell}\right) f((O \setminus Z) \cup G) + \frac{1}{\ell}(f((O \setminus Z) \cup (Z \cap G)) - f(O \cup Z))$$

$$\mathbb{E}\left[f(O \cup A)\right] = \frac{1}{\ell} \sum_{i=1}^{\ell} f(O \cup A_{0,i}) \geq \left(1 - \frac{1}{\ell}\right) f(O \cup G) + \frac{1}{\ell} f\left(O \cup \left(\cup_{i=1}^{\ell} A_{0,i}\right)\right)$$

$$\geq \left(1 - \frac{1}{\ell}\right) f(O \cup G) + \frac{1}{\ell} \left(f(O \cup (Z \cap G)) - f(O \cup Z)\right).$$

Therefore, the last two inequalities in the theorem hold.

**Case** $(O \setminus (G \cup Z)) \cap \{a_1, \ldots, a_\ell\} \neq \emptyset$. Then $o_{\max} \in \{a_1, \ldots, a_\ell\}$. Suppose that $a_u = o_{\max}$. We analyze that lemma holds with sets $\{A_{u,l}\}_{l=1}^{\ell}$.

Similar to the analysis of the previous case, let $A_{0,l}^{(j)}$ be $A_{0,l}$ after we add $j$ elements into it, $\tau_l^{(j)}$ be $\tau_l$ when we adopt $j$-th elements into $A_{0,l}$, and $\tau_l^{(1)} = M$. By Line 10, it holds that $A_{u,l}^{(1)} = G \cup \{a_u\}$. Then, we can order $O \setminus (G \cup Z \cup \{a_u\}) = \{o_1, o_2, \ldots\}$ such that the first $\ell$ elements are not selected by any set before we get $A_{u,l}^{(1)}$, the next $\ell$ elements are not selected in any set before we get $A_{u,l}^{(2)}$, and so on. Therefore, Inequality 8 also holds in this case, where $A$ is a random set from $\{A_{u,l}\}_{l=1}^{\ell}$.

Since, $a_u \in O$, and $A_{u,l_1} \cap A_{u,l_2} = G \cup \{a_u\}$ for any $1 \leq l_1 \neq l_2 \leq \ell$, it holds that $((O \setminus Z) \cup A_{u,i_1}) \cap ((O \setminus Z) \cup A_{u,i_2}) = O \cup G$. Following the proof of case $(O \setminus (G \cup Z)) \cap \{a_1, \ldots, a_\ell\} = \emptyset$, the last two inequalities still hold in this case.

Overall, since either one of the above cases happens, the theorem holds. $\qquad \square$

### E.2 Pruning Subroutine

PRUNE is used in Alg. 14 to help construct the $(\alpha, \beta)$-guidance set, where $\alpha = 0.377$ and $\beta = 0.46$. In this section, we provide the pseudocode, guarantee and its analysis.

**Lemma E.2.** Suppose PRUNE(Alg. 13) is called with $(f, \mathcal{A})$ and returns the set $\mathcal{A}'$. For every $A \in \mathcal{A}$ and its related output set $A' \in \mathcal{A}'$, it holds that,

$$1)\ f(A') \geq f(A);$$
$$2)\ f(S \cup A') \geq f(S \cup A), \forall S \subseteq \mathcal{U};$$
$$3)\ f(T) \leq f(A'), \forall T \subseteq A'.$$

**Algorithm 13:** A pruning algorithm which deletes element with negative marginal gain

---

1 **Procedure** PRUNE $(f, \mathcal{A})$**:**
2     **Input:** oracle $f$, a sequence of subsets $\mathcal{A}$
3     Initialize: $\mathcal{A}' \leftarrow \emptyset$
4     **for** $A \in \mathcal{A}$ **do**
5         **for** $x \in A$ **do**
6             **if** $\Delta(x|A - x) < 0$ **then** $A \leftarrow A - x$
7         **end**
8         $\mathcal{A}' \leftarrow \mathcal{A}' \cup \{A\}$
9     **end**
10     **return** $\mathcal{A}'$

---

*Proof.* Let $A_i$ be $A$ after we delete $i$-th element $x_i$, $A_0$ be the input set $A$, $A_m$ be the output set $A'$. Since any element $x_i$ being deleted follows that $\Delta(x_i|A_i) < 0$, it holds that $f(A_i) > f(A_i + x_i) = f(A_{i-1})$. Therefore, $f(A') = f(A_m) > \ldots > f(A_0) = f(A)$. The first inequality holds.

For any $x_i \in A \setminus A'$, it holds that $\Delta(x_i|A_i) < 0$. By submodularity,

$$f(S \cup A) - f(S \cup A') = \sum_{i=1}^{m} \Delta(x_i|S \cup A_i) \leq \sum_{i=1}^{m} \Delta(x_i|A_i) < 0.$$

The second inequality holds.

For any $y \in A' \setminus T$, since it is not deleted, there exists $0 \leq i_y \leq m$ such that $\Delta(y|A_{i_y}) \geq 0$. By submodularity,

$$f(A') - f(T) \geq \sum_{y \in A' \setminus T} \Delta(y|A') \geq \sum_{y \in A' \setminus T} \Delta(y|A_{i_y}) \geq 0.$$

The third inequality holds. $\qquad\qquad\square$

### E.3   Proofs for Theorem 3.1 of Alg. 14

Prior to delving into the proof of Theorem 3.1, we provide the following corollary first. It demonstrates the progression of the intermediate solution of THRESHGUIDEDIG, after the pruning process, relying on Lemma E.2 and E.1.

**Corollary E.1.** Let $O \subseteq \mathcal{U}$ be any set of size at most $k$. Then PRUNE(THRESHGUIDEDIG$(f, k, \emptyset, G, \ell)$) outputs $\ell(\ell + 1)$ candidate sets with $\mathcal{O}\left(\ell^2 n \log(k)/\varepsilon\right)$ queries. Moreover, with a probability of $(\ell + 1)^{-1}$, a randomly selected set $A$ from the output satisfies that:

$$1) \left(\frac{\ell}{1 - \varepsilon} + 1\right) \mathbb{E}[f(A)] \geq \left(1 - \frac{1}{\ell}\right) f(O \cup G) + \frac{\ell}{1 - \varepsilon} f(G) - \varepsilon f(O);$$

$$2) \mathbb{E}[f(O \cup A)] \geq \left(1 - \frac{1}{\ell}\right) f(O \cup G);$$

$$3) f(O \cap A) \leq f(A).$$

**Theorem 3.1.** Let $(f, k)$ be an instance of SM, with the optimal solution set $O$. Algorithm 14 achieves a deterministic $(0.377 - \varepsilon)$ approximation ratio with $\mathcal{O}(n \log(k) \ell_1{}^{2\ell_1} \ell_2{}^{2\ell_2 - 1})$ queries, where $\ell_1 = \frac{10}{3\varepsilon}$ and $\ell_2 = \frac{5}{\varepsilon}$.

*Proof.* Following the proof of Theorem 3.1, we consider two cases of the algorithm.

**Case 1.** For every $A \in Z$, it holds that $f(O \cup A) \geq 0.46 f(O)$. Then, we prove that $\max_{C \in Z} f(C) \geq (0.377 - \varepsilon) f(O)$.

In the following, we prove the theorem by analyzing the random case of the algorithm, where we randomly select a set from the output of PRUNE (THRESHGUIDEDIG). Suppose that we successfullly select a set where the inequalities in Corollary E.1 hold. Let $A_i$ and $A_{i-1}$ be random sets in $Z_i$

**Algorithm 14:** Nearly linear-time deterministic algorithm for size constraint

---

1 **Input:** oracle $f$, size constraint $k$, error rate $\varepsilon$
2 **Initialize** $\varepsilon' = \frac{2}{\varepsilon}, \ell_1 \leftarrow \frac{5}{3\varepsilon'}, \ell_2 \leftarrow \frac{5}{2\varepsilon'}, t \leftarrow 0.3$
3 ▷ *Create guided sets*
4 $Z_0 \leftarrow \emptyset$
5 **for** $i \leftarrow 1$ *to* $\ell_1$ **do**
6      $Z_i \leftarrow \emptyset$
7      **for** $A_{i-1} \in Z_{i-1}$ **do**
8          $Z_i \leftarrow Z_i \cup \text{PRUNE}(\text{THRESHGUIDEDIG}(f, k, \emptyset, A_{i-1}, \ell_1, \varepsilon'))$
9      **end**
10 **end**
11 $Z \leftarrow \cup_{i=1}^{\ell_1} Z_i$
12 ▷ *Build solution based on guided sets*
13 $G \leftarrow \emptyset$
14 **for** $A \in Z$ **do**
15      $G_0 \leftarrow \emptyset$
16      **for** $i \leftarrow 1$ *to* $\ell_2$ **do**
17          $G_i \leftarrow \emptyset$
18          **for** $B_{i-1} \in G_{i-1}$ **do**
19              **if** $i \leq t\ell$ **then**
20                  $G_i \leftarrow G_i \cup \text{THRESHGUIDEDIG}(f, k, A, B_{i-1}, \ell_2, \varepsilon')$
21              **else**
22                  $G_i \leftarrow G_i \cup \text{THRESHGUIDEDIG}(f, k, \emptyset, B_{i-1}, \ell_2, \varepsilon')$
23              **end**
24          **end**
25      **end**
26      $G \leftarrow G \cup G_{\ell_2}$
27 **end**
28 **return** $C^* \leftarrow \arg\max_{C \in Z \cup G} f(C)$

---

and $Z_{i-1}$, respectively. Let $\left(1 - \frac{1}{\ell_1}\right)^{i^*-1} > 0.46 \geq \left(1 - \frac{1}{\ell_1}\right)^{i^*}$. Then, by Inequality (2) in Corollary E.1, when $i < i^*$, $\mathbb{E}\left[f(O \cup A_i)\right] \geq \left(1 - \frac{1}{\ell_1}\right)^i f(O)$; when $i \geq i^*$, $f(O \cup A_i) \geq 0.46 f(O)$ by assumption. By applying Inequality (1) in Corollary E.1,

$$\mathbb{E}\left[f\left(A_{i^*}\right)\right] \geq \left[\frac{i^*}{\frac{\ell_1}{1-\varepsilon'}+1}\left(1 - \frac{1}{\ell_1}\right)^{i^*} - \varepsilon'\left(1 - \left(1 - \frac{1}{\frac{\ell_1}{1-\varepsilon'}+1}\right)^{i^*}\right)\right] f(O)$$

$$\mathbb{E}\left[f\left(A_{\ell_1}\right)\right] \geq \left(1 - \frac{1}{\frac{\ell_1}{1-\varepsilon'}+1}\right)^{\ell_1 - i^*} \mathbb{E}\left[f\left(A_{i^*}\right)\right] + \left(1 - \left(1 - \frac{1}{\frac{\ell_1}{1-\varepsilon'}+1}\right)^{\ell_1 - i^*}\right)(0.46 - \varepsilon')f(O)$$

$$\geq \left[\frac{i^*}{\frac{\ell_1}{1-\varepsilon'}+1}\left(1 - \frac{1}{\ell_1}\right)^{i^*}\left(1 - \frac{1}{\frac{\ell_1}{1-\varepsilon'}+1}\right)^{\ell_1 - i^*} + \left(1 - \left(1 - \frac{1}{\frac{\ell_1}{1-\varepsilon'}+1}\right)^{\ell_1 - i^*}\right)0.46\right.$$

$$\left. -\varepsilon'\left(1 - \left(1 - \frac{1}{\frac{\ell_1}{1-\varepsilon'}+1}\right)^{\ell_1}\right)\right] f(O)$$

$$\geq \left[\frac{\log(0.46)}{\left(\frac{\ell_1}{1-\varepsilon'}+1\right)\log\left(1 - \frac{1}{\ell_1}\right)}\left(1 - \frac{1}{\ell_1}\right)e^{-1} + \left(1 - \frac{e^{\varepsilon'-1}}{0.46\left(1 - \frac{1}{\frac{\ell_1}{1-\varepsilon'}+1}\right)}\right)0.46 - \varepsilon'\left(1 - e^{-1+\varepsilon'}\right)\right] f(O)$$

$$\text{(Lemma A.4; } \left(1 - \frac{1}{\ell_1}\right)^{i^*-1} > 0.46 \geq \left(1 - \frac{1}{\ell_1}\right)^{i^*})$$

$$\geq (0.377 - \varepsilon)f(O) \qquad\qquad\qquad\qquad (\ell_1 = \tfrac{5}{3\varepsilon'}; \varepsilon' = \tfrac{\varepsilon}{2}; 0 < \varepsilon < 0.377)$$

Since we return the best solution in $Z$ and $G$, it holds that $f(C^*) \geq \mathbb{E}\left[f\left(A_{\ell_1}\right)\right] \geq (0.377 - \varepsilon)f(O)$.

**Case 2.** There exists $A \in Z$, such that $f(O \cup A) < 0.46f(O)$. Then, we prove that $\max_{C \in G} f(C) \geq (0.377 - \varepsilon)f(O)$.

Suppose that $f(A) < 0.377f(O)$. Otherwise, $f(C^*) \geq 0.377f(O)$ immediately. By Inequality (3) in Corollary E.2, it holds that $f(O \cap A) \leq f(A) < 0.377f(O)$. Let $B_{\ell_2}$ be a randomly selected set in $G_{\ell_2}$, where we calculate $G_{\ell_2}$ with the guidance set $A$ and inequalities in Theorem D.2 hold successfully. In the following, we also consider that randomized version of the algorithm, where we randomly select a set $B_i$ from all the solution set returned by THRESHGUIDEDIG. Suppose that we successfully select a set where the inequalities in Lemma E.1 hold. Then, the recursion of $\mathbb{E}\left[f\left(B_i\right)\right]$ can be calculated as follows,

$$\mathbb{E}\left[f\left(B_i\right)\right] \geq \frac{\frac{\ell_2}{1-\varepsilon'}}{\frac{\ell_2}{1-\varepsilon'}+1}\mathbb{E}\left[f\left(B_{i-1}\right)\right] + \frac{1}{\frac{\ell_2}{1-\varepsilon'}+1}\left(f(O \setminus A) - \left(1 - \left(1 - \frac{1}{\ell_2}\right)^i\right)f(O \cup A) - \varepsilon'f(O)\right),$$
$$1 \leq i \leq t\ell_2$$

$$\mathbb{E}\left[f\left(B_i\right)\right] \geq \frac{\frac{\ell_2}{1-\varepsilon'}}{\frac{\ell_2}{1-\varepsilon'}+1}\mathbb{E}\left[f\left(B_{i-1}\right)\right] + \frac{1}{\frac{\ell_2}{1-\varepsilon'}+1}\left[\left(\left(1 - \frac{1}{\ell_2}\right)^{i-\lfloor t\ell_2 \rfloor} - \varepsilon'\right)f(O)\right.$$
$$\left. - \left(\left(1 - \frac{1}{\ell_2}\right)^{i-\lfloor t\ell_2 \rfloor} - \left(1 - \frac{1}{\ell_2}\right)^i\right)f(O \cup A)\right], \qquad\qquad t\ell_2 < i \leq \ell_2$$

Then, by solving the above recursion, it holds that

$$\mathbb{E}\left[f\left(B_{\lfloor t\ell_2 \rfloor}\right)\right] \geq \left(1 - \left(1 - \frac{1}{\ell_2}\right)^{\lfloor t\ell_2 \rfloor}\right)(f(O \setminus A) - f(O \cup A) - \varepsilon'f(O)) + \frac{\lfloor t\ell_2 \rfloor}{\frac{\ell_2}{1-\varepsilon'}+1}\left(1 - \frac{1}{\ell_2}\right)^{\lfloor t\ell_2 \rfloor}f(O \cup A)$$

$$\mathbb{E}\left[f\left(B_{\ell_2}\right)\right] \geq \left(1 - \frac{1}{\ell_2}\right)^{\ell_2 \lfloor t\ell_2 \rfloor}\mathbb{E}\left[f\left(B_{\lfloor t\ell_2 \rfloor}\right)\right] + \frac{\ell_2 \lfloor t\ell_2 \rfloor}{\frac{\ell_2}{1-\varepsilon'}+1}\left(1 - \frac{1}{\ell_2}\right)^{\ell_2 \lfloor t\ell_2 \rfloor}\left[f(O) - \left(1 - \left(1 - \frac{1}{\ell_2}\right)^{\lfloor t\ell_2 \rfloor}\right)f(O \cup A)\right]$$

$$- \frac{\varepsilon'\ell_2}{\frac{\ell_2}{1-\varepsilon'}+1}\left(1 - \left(1 - \frac{1}{\ell_2}\right)^{\ell_2 \lfloor t\ell_2 \rfloor}\right)f(O)$$

$$\geq \left(\left(1 - \frac{1}{\ell_2}\right)^{(1-t)\ell_2+1} - \left(1 - \frac{1}{\ell_2}\right)^{\ell_2}\right)(f(O) - f(O \cap A) - f(O \cup A) - \varepsilon'f(O))$$

$$+ \frac{(1-t)\ell_2}{\frac{\ell_2}{1-\varepsilon'}+1}\left(1 - \frac{1}{\ell_2}\right)^{(1-t)\ell_2+1}f(O) - \frac{\ell_2}{\frac{\ell_2}{1-\varepsilon'}+1}\left((1-t)\left(1 - \frac{1}{\ell_2}\right)^{(1-t)\ell_2} - \left(1 - \frac{1}{\ell_2}\right)^{\ell_2}\right)f(O \cup A)$$

$$- \frac{\varepsilon'\ell_2}{\frac{\ell_2}{1-\varepsilon'}+1}\left(1 - \left(1 - \frac{1}{\ell_2}\right)^{(1-t)\ell_2+1}\right)f(O) \qquad\qquad (t\ell_2 - 1 < \lfloor t\ell_2 \rfloor \leq t\ell_2)$$

$$\geq \left(\left(1 - \frac{1}{\ell_2}\right)^2 e^{t-1} - e^{-1}\right)(f(O) - f(O \cap A) - f(O \cup A) - \varepsilon'f(O))$$

$$+ \frac{(1-t)(\ell_2 - 1)}{\frac{\ell_2}{1-\varepsilon'}+1}\left(1 - \frac{1}{\ell_2}\right)e^{t-1}f(O) - \frac{\ell_2}{\frac{\ell_2}{1-\varepsilon'}+1}\left((1-t)e^{t-1} - \left(1 - \frac{1}{\ell_2}\right)e^{-1}\right)f(O \cup A)$$

$$- \frac{\varepsilon'\ell_2}{\frac{\ell_2}{1-\varepsilon'}+1}\left(1 - \left(1 - \frac{1}{\ell_2}\right)^2 e^{t-1}\right)f(O)$$
$$(f(O \cap A) + f(O \cup A) \geq 0.837f(O); \text{Lemma A.4})$$

$$\geq \left[\left(\frac{(1-t)(\ell_2 - 1)}{\frac{\ell_2}{1-\varepsilon'}+1} + 1 - \frac{1}{\ell_2}\right)\left(1 - \frac{1}{\ell_2}\right)e^{t-1} - e^{-1} - \varepsilon'\left(1 - e^{-1}\right)\right]f(O)$$

$$- \left(e^{t-1} - e^{-1}\right)f(O \cap A) - \left((2-t)e^{t-1} - \left(2 - \frac{1}{\ell_2}\right)e^{-1}\right)f(O \cup A)$$

$$\geq \left[ \left( \frac{(1-t)(\ell_2 - 1)}{\frac{\ell_2}{1-\varepsilon'} + 1} + 1 - \frac{1}{\ell_2} \right) \left( 1 - \frac{1}{\ell_2} \right) e^{t-1} - e^{-1} - \varepsilon' \left( 1 - e^{-1} \right) \right.$$

$$\left. -0.377 \left( e^{t-1} - e^{-1} \right) - 0.46 \left( (2-t)e^{t-1} - \left( 2 - \frac{1}{\ell_2} \right) e^{-1} \right) \right] f(O)$$
$$(f(O \cap A) < 0.377 f(O); f(O \cup A) < 0.46 f(O))$$

$$\geq (0.377 - \varepsilon) f(O) \qquad\qquad (t = 0.3; \ell_2 = \tfrac{5}{2\varepsilon'}; \varepsilon' = \tfrac{\varepsilon}{2})$$

$$\square$$

## F    Experiments

**Experimental setup**    We run all experiments on an Intel Xeon(R) W5-2445 CPU at 3.10 GHz with 20 cores, 64 GB of memory, and one NVIDIA RTX A4000 with 16 GB of memory. For Maximum Cut experiments, we use the standard multiprocessing provided in Python, which takes about 20 minutes to complete, while the video summarization finishes in under a minute.

### F.1    Additional tables and plots

In this section, you can find the tables and plots omitted in the main paper due to space constraints. In Figure 5, we compare the frames selected by FASTLS +RANDOMGREEDY and STANDARDGREEDY, and in Figure 6, we report the results for Barab'asi-Albert and Watts-Strogatz models for Maximum Cut.

### F.2    Problem Formulation

In this section, we formally introduce video summarization and Maximum Cut.

#### F.2.1    Video summarization

Formally, given $n$ frames from a video, we present each frame by a $p$-dimensional vector. Let $X \in \mathcal{R}^{n \times n}$ be the Gramian matrix of the $n$ resulting vectors so $X_{ij}$ quantifies the similarity between two vectors through their inner product. The Determinantal Point Process (DPP) objective function is defined by the determinant function $f : 2^n \to \mathcal{R} : f(S) = \log(\det(X_S) + 1)$, where $X_S$ is the principal submatrix of $X$ indexed by $S$ following Banihashem et al. [1] to make the objective function $f$ a non-monotone non-negative submodular function.

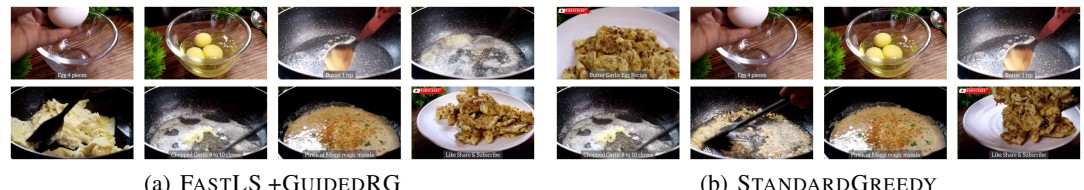

|   (a) FASTLS +GUIDEDRG   |   (b) STANDARDGREEDY   |

Figure 5: Frames selected for Video Summarization

#### F.2.2    Maximum cut

Given an undirected graph $G(V, E)$, where $V$ represents the set of vertices, $E$ denotes the set of edges and weights $w(u, v)$ on the edges $(u, v) \in E$, the goal of the Maximum Cut problem is to find a subset of nodes $S \subseteq V$ that maximizes the objective function, $f(S) = \sum_{u \in S, v \in V \setminus S} w(u, v)$.

### F.3    Hyperparameters

For all experiments, we set the error rate, $\epsilon$, to $0.01$ for FASTLS +GUIDEDRG and to $0.1$ for Lee et al. [21]. Additionally, for video summarization, we run RANDOMGREEDY 20 times and report the standard deviation of these runs. For all other experiments, we run the algorithms once per instance and report the standard deviation over instances.

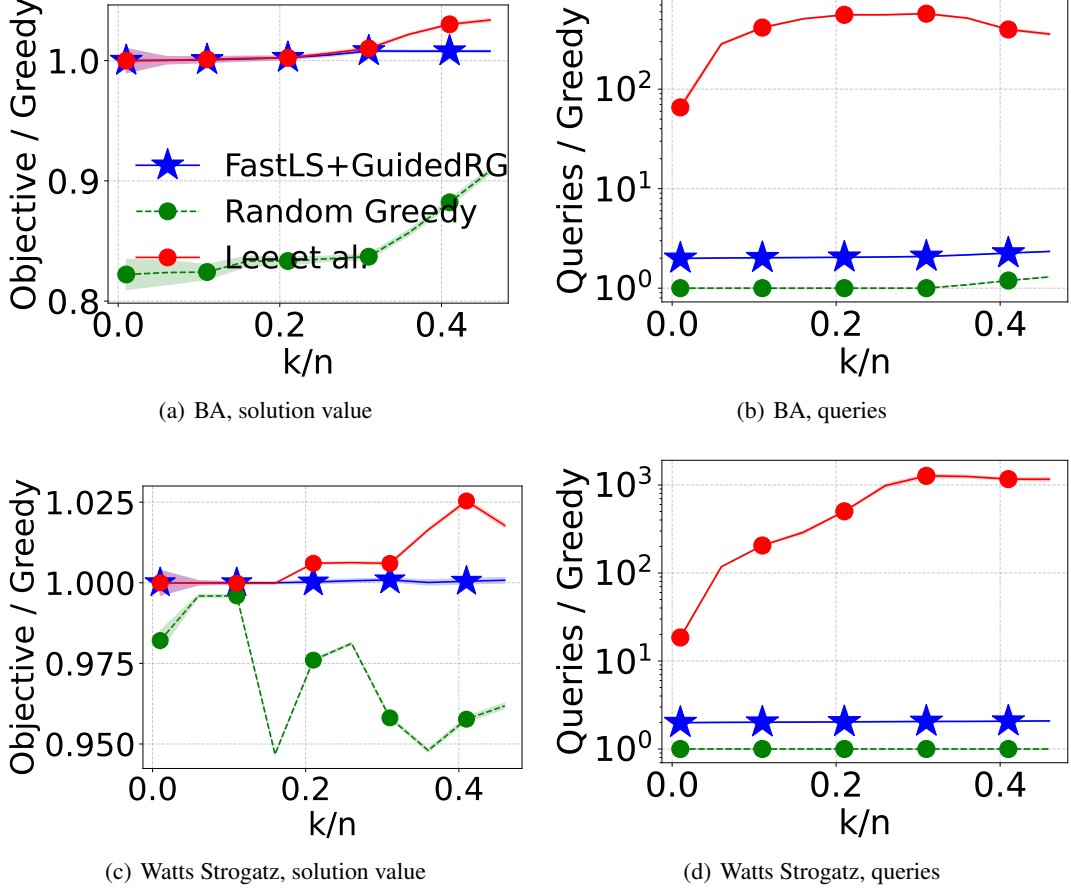

(a) BA, solution value

(b) BA, queries

(c) Watts Strogatz, solution value

(d) Watts Strogatz, queries

Figure 6: The objective value (higher is better) and the number of queries (lower is better) are normalized by those of STANDARDGREEDY. Our algorithm (blue star) outperforms every baseline on at least one of these two metrics.

## F.4 Datasets

The video we select [29] (available under CC BY license) for video summarization lasts for roughly $4$ minutes, and we uniformly sample 100 frames from the video to form the ground set. For maximum cut, we run experiments on synthetic random graphs, each distribution consisting of 20 graphs of size $10,000$ vertices generated using the Erdős-Renyi (ER), Barabasi-Albert (BA), and Watts-Strogatz (WS) models. The ER graphs are generated with $p = 0.001$, while the WS graphs are created with $p = 0.001$ and 10 edges per node. For the BA model, graphs are generated by adding $m = 2$ edges in each iteration. Data and code are provided in the supplementary material to regenerate the empirical results provided in the paper.

## F.5 Implementation of FASTLS+GUIDEDRG

For our implementation of FASTLS, we take the solution of STANDARDGREEDY as our initial solution $Z_0$; the theoretical guarantee is thus $f(Z_0) > \text{OPT}/k$, since $Z_0$ has higher $f$-value than the maximum singleton. This increases the theoretical query complexity of our algorithm as implemented to $\mathcal{O}\left(\frac{kn}{\varepsilon}\log(\frac{k}{\varepsilon})\right)$. Then, for each swap, we find the best candidates to remove from and add to the current solution set (including the dummy element), rather than any pair that satisfies the criterion. For guided GUIDEDRG, we implement exactly as in the pseudocode (Alg. 6) – we remark that we could instead use (a guided version of) the linear-time variant of RANDOMGREEDY [10] to reduce the empirical number of queries further, but for simplicity we did not do this in our evaluation.

