# OpenReview forum: "Discretely beyond $1/e$: Guided Combinatorial Algortihms for Submodular Maximization"
_NeurIPS.cc/2024/Conference — NeurIPS 2024 poster_

### Official Review · Reviewer_arzS · 2024-07-10

**Soundness:** 3
**Presentation:** 3
**Contribution:** 4
**Rating:** 7
**Confidence:** 4

**Summary:**

This paper studies the fundamental problem of improving the approximation factor for non-monotone submodular maximization subject to cardinality or matroid constraints. In particular, the focus of the work is on designing combinatorial algorithms as opposed to continuous methods based on the multilinear extension.

The paper has the following contributions:
for cardinality constraints:
- a combinatorial $0.385 - \varepsilon$ approximation is proposed, which runs in $O(nk/\varepsilon)$. The algorithm is randomized but can be derandomized with a multiplicative deterioration of the query complexity which is exponential in $1/\varepsilon$
- a deterministic $0.377 - \varepsilon$ approximation is obtained, with running time $O(n \log k\cdot C_{\varepsilon})$, where $C_{\varepsilon}$ is exponential in $1/\varepsilon$.
- for context, the previous best combinatorial algorithm provided a $0.367$ approximation, while the overall state of the art is a continuous $0.41$-approximation.
for matroid constraints:
- a combinatorial $0.305 - \varepsilon$ approximation is proposed, which runs in $O(nk/\varepsilon)$. The algorithm is randomized but can be derandomized with a multiplicative deterioration of the query complexity, which is exponential in $1/\varepsilon$
- the previous best combinatorial algorithm provided a $0.283$ approximation, while the overall state of the art is a continuous $0.41$-approximation.

**Strengths:**

- Submodular maximization is an important problem to the ICML and NeurIPS  community, with many practical applications and a rich literature in these conferences
- Closing the approximation gap for non-monotone submodular functions is an exciting research question. To get some context, the current state of the art, i.e., the 0.401 approximation by Buchbinder and Feldman, has recently been presented at STOC, one of the top conferences in theoretical computer science
- finding combinatorial algorithms is important as continuous algorithms are impractical to implement and extremely expensive in terms of queries to the submodular function

**Weaknesses:**

the deterministic algorithms are impractical, as they feature an exponential dependence on the approximation parameter $1/\varepsilon$. Their relevance is mainly theoretical.

**Questions:**

None

**Limitations:**

The paper has no potential negative societal impact

---

### Official Review · Reviewer_qSWx · 2024-07-13

**Soundness:** 3
**Presentation:** 3
**Contribution:** 3
**Rating:** 6
**Confidence:** 4

**Summary:**

In this paper, the authors develop a combinatorial algorithm for non-monotone submodular maximization under both size and matroid constraints. Their algorithm uses local search to guide RANDOM GREEDY and INTERPOLATEDGREEDY, providing both a randomized algorithm and a deterministic algorithm. For size constraints, the algorithm achieves an approximation ratio of 0.385, and for matroid constraints, it achieves an approximation ratio of 0.305. It is the first combinatorial algorithm with an approximation ratio greater than 1/e for size constraints. The technique in this paper is similar to that in [6]. However, while the algorithm in [6] is a continuous algorithm requiring a query complexity of $\Omega(n^{10}) $, the algorithm presented in this paper only requires $O(kn)$ queries of a value oracle. They also propose another algorithm based on this approach, which replaces the greedy part in INTERPOLATEDGREEDY with the descending thresholds technique. This algorithm achieves an approximation ratio of 0.377 for size constraints in nearly linear query time $O(k \log n)$, which is slightly greater than 1/e.

**Strengths:**

1. The paper provides two new deterministic algorithms with approximation ratio better than 1/e.
2. The paper provides practical randomized algorithm with both theoretical analysis and experimental evaluation.

**Weaknesses:**

1.	The results in the paper are not strong enough. For the randomized algorithm, the paper an algorithm with query complexity $O(kn/\epsilon)$ and approximation ratio 0.385. It is worse than the most recent paper (https://arxiv.org/abs/2405.13994). For two deterministic algorithms, they are impractical since the dependence of $\epsilon$ is too large.

**Questions:**

Both the results and techniques in this paper are somewhat similar to those in the recent paper (https://arxiv.org/abs/2405.13994). It might be beneficial to explain the relationship between your paper and theirs, especially the difference in the techinique ideas.

**Limitations:**

Yes

---

> ### Author Rebuttal · Authors · 2024-08-05
>
> > Both the results and techniques in this paper are somewhat similar to those in the recent paper (https://arxiv.org/abs/2405.13994). It might be beneficial to explain the relationship between your paper and theirs, especially the difference in the techinique ideas.
>
> Thank you for bringing this paper to our attention; we note that
>   it was released on arXiv after the submission deadline
>   for NeurIPS 2024. Also, we would like to respectfully point out the
>   policy of NeurIPS concerning contemporaneous work.
>
> Indeed, the authors of arXiv:2405.13994 had a similar goal to our work,
>   namely to produce practical algorithms with guarantee larger than $1/e$.
>   It appears to be a case of independent, parallel work.
>   Broadly speaking, the main idea of the two works is similar -- to develop
>   methods to guide the random greedy algorithm with a fast algorithm to
>   find a local optimum.
>   Our local search procedure is deterministic, while theirs is randomized
>   and uses different ideas. Their local search
>   is faster for small $k$ values. They also guide a faster variant of
>   RandomGreedy to achieve their overall time complexity of
>   $O(n + k^2)$.
>
> On the other hand,
>   in our work, we 1) develop a guided algorithm
>   for matroid constraints;  2) for size constraint,
>   we have an asymptotically faster algorithm that uses a novel
>   way of guiding with partial solutions from random greedy itself,
>   which are not local optima, thereby achieving ratio $0.377$ in $O_{\epsilon}(n \log k)$;
>   and 3) we derandomize.
>   For 1) and 2), we had to develop novel recurrences to analyze
>   random greedy, which depart further from the
>   original guiding and analysis of the continuous measured greedy algorithm
>   to achieve ratio $0.385$. Random greedy for matroids is much
>   more like a local search algorithm than a greedy algorithm,
>   as the best swap is made at each iteration.
>   In particular, the guided recurrences stated in Lemmas C.5 and C.6
>   for the matroid constraint, and their closed form solution in
>   Lemma C.7, as illustrated in Fig. 4.
>   Further, derandomization for the matroid constraint
>   required development of Algorithm 8 and its analysis in
>   Lemma D.1. For 2), we think the use of random greedy to
>   guide itself is interesting, and we believe the resulting
>   analysis (depicted in Fig. 2) aids in the understanding
>   of the random greedy algorithm, perhaps opening up further
>   uses such as bicriteria algorithms, etc. Although the deterministic version has poor dependence on $\epsilon$, this idea might be used to produce a faster randomized version of this algorithm in subsequent work.
>
> In future versions of our paper, we will add a citation to this independent, parallel work
>   and a discussion of the technical relationship.

---

> > ### Comment · Reviewer_95ga · 2024-08-08
> > **response to contemporaneous work**
> >
> > I want to echo the authors on this point. I do not think it is appropriate to compare their results with results from work that was clearly done in parallel. Here is the NeurIPS policy on contemporaneous work.
> >
> > **Contemporaneous Work**: For the purpose of the reviewing process, papers that appeared online within two months of a submission will generally be considered "contemporaneous" in the sense that the submission will not be rejected on the basis of the comparison to contemporaneous work. Authors are still expected to cite and discuss contemporaneous work and perform empirical comparisons to the degree feasible. Any paper that influenced the submission is considered prior work and must be cited and discussed as such. Submissions that are very similar to contemporaneous work will undergo additional scrutiny to prevent cases of plagiarism and missing credit to prior work.

---

> > ### Comment · Reviewer_qSWx · 2024-08-13
> >
> > Thank you for your response. I've increase my score.

---

### Official Review · Reviewer_95ga · 2024-07-16

**Soundness:** 3
**Presentation:** 1
**Contribution:** 3
**Rating:** 6
**Confidence:** 4

**Summary:**

This paper studies the classical problem of maximizing a non-monotone submodular function subject to a cardinality and a matroid constraint.
A long line of work has developed approximation algorithms for these problems which achieve a 0.401approximation; however, every algorithm which achieves better than $1/e = 0.368$ relies on the multilinear extension and is primarily of theoretical interest, as the run times are very large (but polynomial).
The more practical combinatorial algorithms have not been able to break the $1/e$ barrier.

The main contribution is a suite of algorithms which work via an "guided" local search procedure.
The main algorithm works as follows: a local search algorithm obtains a first solution and then an "guided" local search uses this first solution to obtain a second one.
If the parameters of the algorithm is appropriately set, at least one of these two solutions is of high (expected) quality.
The main result is that this (randomized) algorithm achieves a $0.385-\epsilon$-approximation for cardinality constraint and $0.305-\epsilon$ approximation for matroid constraint and runs in $\mathcal{O}(n k / \epsilon)$ time.

There are a variety of other results, including several de-randomizations of this algorithm, though those are mostly of theoretical interest because their run time is exponential in $\epsilon$.
Simulations are run comparing the algorithms to two reasonable benchmarks.

**Strengths:**

The main strength of the work is in the development of simple algorithms with improved approximation ratios for submodular maximization.
In particular, the randomized algorithm achieving $0.385-\epsilon$-approximation for cardinality constraint and $0.305-\epsilon$ approximation for matroid constraint is the main contribution of the paper.
Of particular note is that this algorithm is analyzed for both cardinality and matroid constraints, though different switching times $t$ should be used.

I really enjoyed the clarity of the writing in Section 2.2 which provided intuition for the theoretical results.

**Weaknesses:**

**Paper Clarity**: For the most part, the paper is not well organized and can be challenging to read. With the exception of Section 2.2, the writing is a bit rushed and lacking focus. Many times I had to go back and forth in the paper just to read it. Perhaps the reason for this is that several algorithms are being presented with various guarantees and settings. While the results are great, the paper requires some serious efforts in improving the readability.

**Simulations**: While I appreciate the authors' work in constructing simulations, they are admittedly quite weak or at the very least poorly explained. This may be due in large part to space constraints. I summarize the issues here:

- The objective functions are not presented and the movie summarization dataset is not discussed.
- It is claimed that max-cut is run on 3 different random graph models -- but then how does only one line appear in the Figure 3c?
- If FastLS + GuidedRG is random, then why does it not also have some uncertainty quantification (i.e. standard error bars) on its objective value and run time?
- The simulations are weak because in all of these instances, the more sophisticated algorithms provide very little improvement over the greedy algorithm. I'm sympathetic to this phenomenon, I know it can happen. But then I think that the experiments should focus on practical instances where the sophisticated algorithms do actually improve over the greedy algorithm.

**Questions:**

I thank the authors for their manuscript and invite them to respond to the points raised in my review.
However, I do not feel that I have any questions to raise at this time.

**Limitations:**

All limitations sufficiently addressed.

---

> ### Author Rebuttal · Authors · 2024-08-05
>
> We thank the reviewer for the thoughtful feedback. We respond to each point below.
> > Paper Clarity: For the most part, the paper is not well organized and can be challenging to read. With the exception of Section 2.2, the writing is a bit rushed and lacking focus. Many times I had to go back and forth in the paper just to read it. Perhaps the reason for this is that several algorithms are being presented with various guarantees and settings. While the results are great, the paper requires some serious efforts in improving the readability.
>
> We apologize for the difficulty. Due to space constraints, it was difficult to decide what
>   should be presented in the main text vs. the appendices. We decided 1) to limit the main
>   text to the size constraint, as it is more accessible than the matroid constraint,
>   and 2) to try to explain the most important ideas in the main text at a high level, and
>   leave the formal analysis to the appendix. However, this means we sometimes refer to
>   lemmas, etc., in the appendix, which decreases readability. In the next revision, we will
>   make a further refinement in the hope of increasing readibility further and to minimize
>   the necessity of consulting back and forth between the appendices and main text.
>
> > The objective functions are not presented and the movie summarization dataset is not discussed.
>
> Due to space constraints, the objective functions for each application,
> and the movie summarization dataset details are discussed in Appendix F.
> We will make this more clear in the next version.
>
> > It is claimed that max-cut is run on 3 different random graph models -- but then how does only one line appear in the Figure 3c?
>
> In the main text, we only present the results on ER graphs.
> The results on the other two models are provided in Fig. 6 in Appendix F.
> We will add references in the experiment section to clarify.
>
> > If FastLS + GuidedRG is random, then why does it not also have some uncertainty quantification (i.e. standard error bars) on its objective value and run time?
>
> Thank you for asking this. Interestingly, it turns out that the objective value of our algorithm is usually dominated by
> the local search subroutine, which is deterministic. Thus, there is little variance
> in the objective value, which is why the error bars are not visible (they are there).
> For the runtime, the number of queries of GuidedRG is deterministic.
> Thus, the number of queries of our main algorithm is deterministic.
>
> > The simulations are weak because in all of these instances, the more sophisticated algorithms provide very little improvement over the greedy algorithm. I'm sympathetic to this phenomenon, I know it can happen. But then I think that the experiments should focus on practical instances where the sophisticated algorithms do actually improve over the greedy algorithm.
>
> In our results, we observe an improvement over Greedy of up to 2\%,
> which in our opinion is significant.
> However, we agree that we could have chosen applications, perhaps weakly submodular ones,
> in which further improvement on Greedy is possible. Analyzing our algorithms in the weakly submodular case both theoretically and empirically is an interesting avenue for future work.
>
> In addition, Greedy does not guarantee a constant approximation ratio for non-monotone submodular maximization problems.
> Compared to those algorithms which do provide constant approximation ratio,
> our algorithm outperforms RandomGreedy with respect to the objective value,
> and outperforms Lee et al.'s LocalSearch with respect to the query complexity.

---

> > ### Comment · Reviewer_95ga · 2024-08-08
> > **response to authors**
> >
> > I thank the authors for their thoughtful response to my review.
> >
> > Their responses on the simulations have been satisfactory. I understand and agree with all points (except perhaps that 2% increase is significant, but we don’t have to discuss this further). I believe they will be addressed in a revision.
> >
> > I understand that the space constraints make writing the paper difficult. I am not opposed to the idea of putting the size constraint in the main body and the matroid, together with all formal analysis in the appendix
> >
> > But let me be a bit more specific about the issue of clarity. The way I see it, the main algorithm of the paper is Algorithm 2. However, when the reader reads the pseudocode of Algorithm 2 on top of page 4, it conveys very little meaning about the algorithm itself. Namely, GuidedRG is not defined nearby in the paper (indeed, it is only defined in the appendix). So, as the reader, it is quite confusing to read Section 2.0 as the algorithms aren’t defined sufficiently well for us to have an understanding yet. A similar thing happens in Definitions 2.1 and Lemma 2.2 — these are outputs of an algorithm which is still unfamiliar to the reader at that point. In this way, the reader is having to skip back and forth between parts of the paper upon a first read, which does detract from the clarity. You might consider shortening or moving Sec 2.3 and 3 to the appendix to describe GuidedRG in the main paper.
> >
> > I mention this only to be more specific about what I meant by paper clarity and offer suggestions for improvement.
> >
> > I like the theoretical results quite a lot and I think that authors are capable of revising the paper so as to improve clarity. The only difficult is that such revisions appear to require non-trivial re-organization of the paper. As a reviewer, this puts me in a tough place. I am uneasy with significantly raising my score because the authors could revise the paper to make it clearer. We should be reviewing the submitted paper rather than the paper that could be resubmitted. This is especially true given that there is no way to accept only after verifying the appropriate revisions have been made. On the other hand, it would be a shame for the paper to not appear in this conference, given its strong technical results. I would also feel bad if rejection of this paper meant it would no longer be viewed as contemporary to Tukan et al (2024).
> >
> > I will raise my score to weak accept to signal to the AC that I am supportive of the technical results in the paper, but still have concerns about the presentation.

---

> > > ### Author Response · Authors · 2024-08-11
> > > **response**
> > >
> > > Thank you for the feedback and for the specific example concerning the presentation. We agree that our style of introducing the final algorithm or result first, before defining each piece, is difficult to read. However, we believe we can significantly improve the readability without a large amount of reorganization. For example, when introducing Alg. 2, we can rephrase the description to give a better sense of what each component is, and also that their precise description is deferred:
> > >
> > > > In overview, Alg. 2 consists of two components, which are detailed below. The first component is a local search algorithm described in detail in Section 2.1. In brief, the local search starts from a constant-factor solution and efficiently improves it to nearly a local optimum. The second component is a random greedy algorithm that is guided by the output of the local search, described in detail in Section 2.2...
> > >
> > > In any case, thank you again for the feedback, and we will improve the readability in the next version.

---

### Official Review · Reviewer_NXWY · 2024-07-17

**Soundness:** 3
**Presentation:** 3
**Contribution:** 2
**Rating:** 6
**Confidence:** 3

**Summary:**

The paper investigates combinatorial approximation algorithms for constrained submodular maximization problems. Observing the algorithms that pass the $1/e$ threshold are generally carries the problem to the continuous domain, they investigate the answer to the following question: "Is it possible to obtain approximation ratios that are better than $1/e$ for constrained submodular maximization problems with combinatorial algorithms?" They answer this question affirmatively by combining Buchinder et al.'s RandomGreedy algorithm with a novel fast local search algorithm. Additionally, they propose derandomized versions of these algorithms by altering the InterpolatedGreedy algorithm by Chen and Kuhnle.

**Strengths:**

The paper improves the known existing approximation ratios without resorting to continuous algorithms while using the same number of queries. More specifically, they improve the combinatorial approximation ratios of Buchbinder et al. from $1/e \approx 0.367$ to $0.385 - \varepsilon \approx 1/e + 0.018$ for the cardinality constraint and from $0.283$ to $0.305 - \varepsilon$ for matroid constraints while still requiring $\mathcal{O}(kn)$ queries. The idea of guiding the RandomgGreedy algorithm with the Fast Local Search (FastLS) is original. The ideas are conveyed clearly and the improvement over the state of the art for combinatorial algorithms is quantifiable.

**Weaknesses:**

- On lines 73-75, you mention "Unfortunately, there is no known method to derandomize continuous algorithms, as the only known way to approximate the multilinear extension relies on random sampling methods." You may want to revise this sentence because as far as I know, there is a work that proposes estimating the multilinear relaxation of coverage-like functions with Taylor series expansions instead of sampling. Citation: Özcan, Gözde, Armin Moharrer, and Stratis Ioannidis. "Submodular maximization via Taylor series approximation." Proceedings of the 2021 SIAM International Conference on Data Mining (SDM). Society for Industrial and Applied Mathematics, 2021.

- On line 283, you mention an InterlaceGreedy algorithm for the first time as far as I can tell. Is this a typo or a different algorithm? If it is a different algorithm, where it is described?

**Questions:**

- On lines 114-115, you mention using an LP method for derandomizing the RandomGreedy algorithm and how this method did not work in this case. Could you please elaborate more on why this was the case? Was the problem asymptotic query complexity or something else?

- Why the dummy elements are needed?

-

**Limitations:**

Both in Sections 1.1 and 5, the authors discuss the limitations of their work fairly.

---

> ### Author Rebuttal · Authors · 2024-08-05
>
> We thank the reviewer for the constructive comments.
> > On lines 73-75, you mention "Unfortunately, there is no known method to derandomize continuous algorithms, as the only known way to approximate the multilinear extension relies on random sampling methods." You may want to revise this sentence because as far as I know, there is a work that proposes estimating the multilinear relaxation of coverage-like functions with Taylor series expansions instead of sampling. Citation: Özcan, Gözde, Armin Moharrer, and Stratis Ioannidis. "Submodular maximization via Taylor series approximation." Proceedings of the 2021 SIAM International Conference on Data Mining (SDM). Society for Industrial and Applied Mathematics, 2021.
>
> We thank the reviewer for pointing us to this work, which provides an interesting way to approximate the multilinear extension in the case of coverage functions. We will add it to the discussion, and modify the claim as the reviewer suggests.
>
> > On line 283, you mention an InterlaceGreedy algorithm for the first time as far as I can tell. Is this a typo or a different algorithm? If it is a different algorithm, where it is described?
>
> We apologize for the confusion. InterlaceGreedy is a subroutine of the InterpolatedGreedy of [11] (number 11 of the bibliography in the manuscript). We develop guided versions of it in Alg. 8 and 9 of the appendix.
>
> > On lines 114-115, you mention using an LP method for derandomizing the RandomGreedy algorithm and how this method did not work in this case. Could you please elaborate more on why this was the case? Was the problem asymptotic query complexity or something else?
>
> The LP method relies on a standard lemma for non-monotone submodular functions,
>       which needs a bound on the probability that any element is chosen into the solution.
>       Specifically, Lemma 2.2 of [A] and its generalization in [B]. When the algorithm switches back from the guided
>       behavior to unguided, we were only able to provide a loose upper bound on this probability
>       that wasn't good enough for the analysis. In particular, we had difficulty ordering the
>       the probabilities of the elements in a way that would allow us to apply the lemma.
>       It is possible that a generalization of Lemma 2.2, or a clever way of considering
>       the probabilities is possible and that the LP method could be used.
>
> [A]: Feige et al. Maximizing Non-monotone Submodular Functions. SIAM Journal on Computing 40 (4).
>
>  [B]: Buchbinder et al. Submodular Maximization with Cardinality Constraints. SODA 2014.
>
> > Why are the dummy elements needed?
>
>  We introduce dummy elements to simplify both the pseudocodes and the analysis.
>       The simplification comes from the fact that we may assume that an optimal
>       solution is of size $k$, or a base of the matroid, by adding dummy elements.
>       Moreover, our set $M_i$ in RandomGreedy
>       of the top $k$ marginal gains has $k$ non-negative gains. Otherwise, we would
>       have to allow $M_i$ to potentially be smaller and adjust the probabilities of elements,
>       for example: choose no element with some probability, and one of the remainder
>       uniformly randomly. This would create more cases in the analysis and complicate the
>       picture unnecessarily.

---

### Author Rebuttal · Authors · 2024-08-05

We thank all reviewers for the contructive comments. We hope that we were able to answer the questions of all reviewers in our individual responses. To summarize,
in the next version,
- we will make efforts to improve the readability of the paper by minimizing the necessity of consulting the appendix to understand the theoretical results.
- In the experimental section, we will add more pointers to the appendix to clarify settings and results that are omitted from the main text for space reasons.
- We will add citation of the contemporaneous work [1].
We note that [1] appeared on arXiv on 22 May 2024,
20:56 UTC, after the NeurIPS 2024 submission deadline,
at 22 May 2024, 20:00 UTC -- thus, there was no way for
us to address this paper in the submitted manuscript.
We discuss in detail the differences between the two works in our response to Reviewer qSWx, and a version of this discussion will be added to the paper.

[1]: Tukan et al. Practical 0.385-Approximation for Submodular Maximization Subject to a Cardinality Constraint. arXiv:2405.13994

---

### Decision · Program_Chairs · 2024-09-25

**Decision:**

Accept (poster)

**Comment:**

The main contribution of the paper are combinatorial algorithms for constrained submodular maximization that achieve an approximation guarantee that is better than 1/e.  Several algorithms have been proposed that achieve such a guarantee, but they are continuous algorithms based on the multilinear extension relaxation and are generally inefficient in practice. In contrast, the proposed algorithms are combinatorial and thus have the potential to be much more practical. There was consensus among the reviewers that this is a strong contribution to this classical problem. One of the reviewers noted several issues with the paper's presentation and the authors promised to address them in the revised version.